# The contribution of temporal coding to odor coding and odor perception in humans

Ofer Perl[1], Nahum Nahum[2], Katya Belelovsky[1], Rafi Haddad[1]*

[1]The Gonda Multidisciplinary Brain Research Center, Bar-Ilan University, Ramat-Gan, Israel; [2]Department of Psychology, Bar-Ilan University, Ramat-Gan, Israel

**Abstract** Whether neurons encode information through their spike rates, their activity times or both is an ongoing debate in systems neuroscience. Here, we tested whether humans can discriminate between a pair of temporal odor mixtures (TOMs) composed of the same two components delivered in rapid succession in either one temporal order or its reverse. These TOMs presumably activate the same olfactory neurons but at different times and thus differ mainly in the time of neuron activation. We found that most participants could hardly discriminate between TOMs, although they easily discriminated between a TOM and one of its components. By contrast, participants succeeded in discriminating between the TOMs when they were notified of their successive nature in advance. We thus suggest that the time of glomerulus activation can be exploited to extract odor-related information, although it does not change the odor perception substantially, as should be expected from an odor code per se.

## Introduction

How odors are encoded by the brain is an open fundamental research question in neuroscience, echoing a more general debate regarding how the brain encodes information (*Uchida et al., 2014*). Odor perception starts with odorant molecules binding to olfactory receptors. Different odorants bind to a unique set of possibly overlapping olfactory receptors. In rodents, each olfactory sensory neuron expresses one olfactory sensory receptor and projects to one glomerulus (*Mombaerts, 2006*). Each odorant activates a unique set of glomeruli, which in turn activate the olfactory bulb (OB) output neurons – the mitral and tufted (M/T) cells. These findings led to the hypothesis that odors are encoded by a combinatorial code composed of the set of glomeruli activated by each odor. This code was termed the spatial code or identity code (*Friedrich and Korsching, 1997*; *Friedrich and Korsching, 1998*; *Galizia et al., 1999*; *Johnson and Leon, 2007*; *Malnic et al., 1999*; *Rubin and Katz, 1999*; *Uchida et al., 2000*). Owing to the large number of different glomeruli, this coding scheme is theoretically capable of encoding any number of odors (*Koulakov et al., 2007*).

It has long been observed that odor stimulation evokes odor- and cell-specific temporal patterns of activity in the OB and antennal lobe, which are not directly related to the dynamics of the olfactory stimulus. These observations led to the hypothesis that odors are represented by spatially and temporally distributed ensembles of active neurons. As discussed in detail in *Uchida et al. (2014)*, there are two main models that attempt to describe the role of temporal coding in olfaction: the Hopfield latency coding model and the Laurent slow-evolving decorrelation model.

According to the Hopfield model, odors are encoded by the identity of the odor-activated neurons and their time of activation in relation to some oscillatory cycle (*Figure 1A*) (*Hopfield, 1995*). This coding model provides a simple mechanism for concentration-invariant odor recognition: when the odor concentration changes, multiple neurons shift their spike timing together, such that the relative timing across neurons remains unchanged. According to this model, time is part of the odor

*For correspondence:
rafihaddad@gmail.com

**Competing interests:** The authors declare that no competing interests exist.

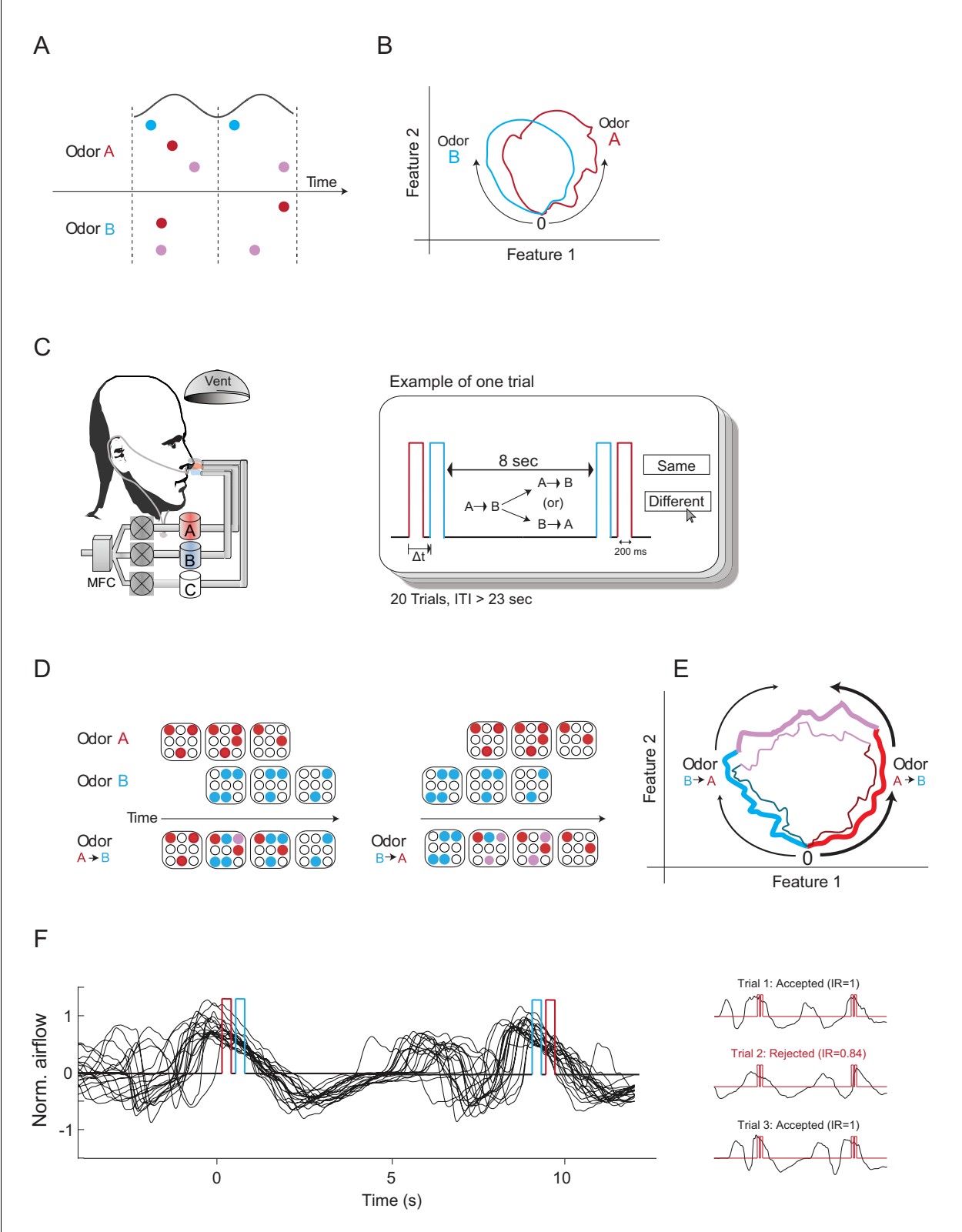

**Figure 1.** Construction of odors differing mainly in their odor-elicited temporal dynamics. (A) A schematic diagram showing three glomeruli (colored circles) responding to two odors over two cycles of a putative oscillatory mechanism (black line). Odors A and B activate the same glomeruli but at different times. According to the Hopfield model, this difference in activation time should be sufficient to render these two odor codes as different odors. (B) Two odor trajectories plotted in phase space. The two trajectories represent two different odors as their trajectories are different in phase

*Figure 1 continued on next page*

*Figure 1 continued*

space. (C) Left panel: experimental setup. Participants were presented with two odors dispensed from two separate canisters using a computer-controlled air dilution olfactometer. Another canister was used to deliver clean air (marked as C). Nasal respiration was recorded using a nasal cannula attached to a pressure-sensitive spirometry sensor. Right panel: experimental design. Two odor pulses were presented consecutively spaced by a time delay (Δt) between them. Eight seconds later, we presented either the same order of odor presentations or a reversed one. Participants then reported whether the odors were same or different. Twenty-three seconds later, this procedure was repeated. Each participant underwent 20 trials. Red, odor A; blue, odor B; ITI, inter-trial-interval. (D) A mockup model of glomeruli responses to two odors delivered one after the other with a short delay in between, both in forward and in reverse order. Red and blue circles represent glomeruli activated by odors A and B, respectively. Purple circles represent glomeruli activated by odors A+B. The order of odor presentation elicits different time sequences of glomeruli activation. (E) Possible trajectories of the two TOMs in phase space. The first trajectory (thick lines) is composed of odor A (red), odors A+B (purple) and then odor B (blue). The second trajectory (thin lines) represents odor B, odor B+A, and odor B. The two trajectories are relatively similar, although the directions are opposite. (F) A typical example of a 20-trial experimental session. Left panel: black traces depict several typical nasal respiration trials overlaid. Positive values represent inhalation. Red and blue triggers represent a single example of odor A and B pulses. Right panel: examples of trials in which the TOMs were presented entirely concurrent (top and bottom) or not concurrent (middle) with sniffing.

The online version of this article includes the following figure supplement(s) for figure 1:

**Figure supplement 1.** Setup validation.

code and therefore two odors that activate a similar set of glomeruli but at different times should evoke two distinguishable percepts. Time can be measured relative to some internal oscillatory cycle such as internal gamma oscillations or the respiration cycle (i.e., phase coding), or relative to inhalation onset regardless of the respiration duration (i.e., latency coding) or relative to the activation times of other neurons (i.e., relative time coding).

Experimental results have provided some support for the Hopfield model. In rodents, the latency of glomerular activation showed stimulus-specific temporal patterns (*Spors, 2006*; *Spors and Grinvald, 2002*). Moreover, increasing the odor concentration reduced the glomeruli onset times while preserving the temporal sequence of glomerular activation, suggesting that the temporal sequence of glomeruli activation encodes the odor identity, whereas the time of activation encodes concentration (*Spors and Grinvald, 2002*). In larvae of *Xenopus laveis* (a model organism that does not sniff), latency from stimulus onset across M/T cells reliably conveyed stimulus information about odor identity and concentration (*Junek et al., 2010*). Thus, these studies show that spike or glomeruli latency with respect to slow oscillations (e.g., respiration cycle), as well as latency from odor onset on the order of hundreds of milliseconds, can convey reliable odor information. Consistent with latency coding, a study reported that downstream neurons in mice are sensitive to the relative activation times of optogenetically activated output neurons in the olfactory bulb (*Haddad et al., 2013*). This finding suggested that cortical neurons can decode specific glomerular activation sequences. Finally, several behavioral studies have shown that mice are able to discriminate between optogenetic stimulations of olfactory neurons, which differ mainly in their activation times, either relative to the sniff cycle or to each other (*Rebello et al., 2014*; *Smear et al., 2011*; *Smear et al., 2013*). These results can be interpreted as behavioral evidence supporting relative- or phase-latency coding as part of the odor code.

A more recent variant of the latency model suggested that odors are encoded by the early components of the activated glomeruli (*Schaefer and Margrie, 2007*; *Wilson et al., 2017*), whereas the components activated later in the evolving spatiotemporal map could possibly be utilized to differentiate between highly similar odors (*Schaefer and Margrie, 2007*). Consistent with this model, a recent study showed that neurons in the piriform cortex tend to respond to the early components and ignore the later ones (*Bolding and Franks, 2018*). Regardless of the exact variant, all of these coding models posit that two odors that differ mainly in terms of their temporal code should elicit a different percept, as time is part of the odor code.

The Laurent model, by contrast, stresses that populations of neurons exhibit synchronized oscillatory activity, but that each neuron only transiently participates in this population activity (*Laurent, 2002*). The identities of neurons that participate in the oscillatory ensemble change over time and form the odor representation that is gradually decorrelated to improve odor encoding in neural space (*Friedrich and Laurent, 2001*; *Friedrich and Laurent, 2004*). Thus, the odor representation can be considered a trajectory in phase space (*Laurent et al., 2001*), where different odors generate different trajectories (*Figure 1B*). Whether the whole trajectory or only parts of it are used for

decoding is unknown. According to this model, temporal dynamics are not a necessary part of the odor code because they only facilitate coding optimization (*Laurent, 2002*). According to one possible interpretation of this model, two odors that activate a similar set of glomeruli but at different times are not necessarily perceived as different, as long as some parts of their evolving trajectories are similar; see *Laurent et al. (2001)* for a full discussion of this model. Support for this model comes from observations of temporally evolving neural dynamics and subsequent analyses demonstrating that the odor representation becomes sparser and more decorrelated over time (*Friedrich and Laurent, 2004*; *Friedrich and Laurent, 2001*; *Gschwend et al., 2015*; *Gupta and Stopfer, 2014*; *Laurent et al., 1996*; *Wehr and Laurent, 1996*) and that disturbing gamma oscillations impairs odor discrimination (*Stopfer et al., 1997*).

Thus overall, the different variants of the latency model predict that two odors that differ substantially in glomerular activation time should be perceived as two different odors as such temporal dynamics are part of the odor code per se. According to the Laurent model, time may act as an orthogonal component to the odor percept, such that two odors that elicit neural responses that differ in their temporal dynamics may be perceived as the same if their neural trajectories resemble one another in some way.

There is still an ongoing debate as to which of these temporal coding models is used by the mammalian olfactory system. Here, we examined whether glomerular activation times affect odor perception in humans. Using temporal odor sequences, we manipulated temporal features that are involved in odor coding and measured their effect on odor perception during discrimination and odor-rating tasks.

## Results

### Construction of odors that differ in their odor-elicited neural temporal dynamics

To test how temporal coding affects odor perception, we manipulated relative glomerular activation times using *temporal odor mixtures* (TOMs). We presented two precise consecutive odor pulses (A and B) separated by a precise short delay ($\Delta t$), and then 8 s later, we presented the same odors again, either in the same order or the reverse order. That is, we first presented TOM $A \underset{\Delta t}{\rightarrow} B$ , and 8 s later, we presented either TOM $B \underset{\Delta t}{\rightarrow} A$ or TOM $A \underset{\Delta t}{\rightarrow} B$ again (*Figure 1C*). Delivering two odors in a different order activates the glomeruli at a different relative time and forms a different latency code (*Figure 1D*), although the odor trajectories in phase space may still share some similar features (*Figure 1E*).

The participants were instructed to decide whether these two consecutively presented TOMs were the same or different. Each odorant was delivered for exactly 200 ms and the second odorant within each TOM was delivered precisely at a controllable, experimentally defined time ($\Delta t$) following the onset of the first odorant (*Figure 1C*). This design ensured that both TOMs had the exact same duration and thus safeguarded against the possible use of odor duration as a cue to facilitate discrimination. Participants were not informed that the TOMs were composed of two odors, to avoid discrimination based on identifying which odor was first (*Laing et al., 1994*).

We verified that odor delivery and clearance were precise across trials with a mini photo-ionizer detector (PID) (miniPID, Aurora Scientific). The average latency of the PID-reported odor signal onset was within the range of 20–40 ms, and the odor concentration returned to 10% of its peak value within 30–50 ms after odor offset (*Figure 1—figure supplement 1A–B*). The time of glomerulus activation depends on odorant-receptor dynamics, among other factors. That said, even assuming some delay in glomerulus activation due to these interactions, switching the order of the odors in the TOM will nevertheless create a substantial difference in the glomerular activation times, because the glomerular activation times of two odors in the first TOM were shifted by $+\Delta t$ and by $-\Delta t$ in the second TOM. Thus, our two TOMs were expected to activate a similar set of glomeruli, but the sequence of glomerular activation time of each TOM was substantially different.

We used an inter-stimulus-interval (ISI) of 8 s between the two TOMs rather than the more common 10–20 s interval employed in human olfaction experimentation, because we observed in pilot sessions that when ISI was longer, performance was significantly hampered even when the

participants were asked to discriminate between the two constituent odors (i.e., odor A vs. odor B). This was probably due to the fact that the task essentially became an olfactory match-to-sample working memory task rather than a discrimination task, which is more difficult as the ISI gets longer (*Arzi et al., 2014*; *Zelano et al., 2009*). An inter-trial-interval (ITI) of at least 23 s separated discrimination trials, depending on the participants' response times in the trial (*Figure 1C*). To minimize sensory fatigue resulting from repeated odor exposure, each participant conducted a maximum of 20 trials (i.e., 20 × 2 = 40 odor trials), which is within the range used in olfaction psychophysics experiments (*Johnson et al., 2003*; *Weiss et al., 2012*).

Thus, this experimental setup allowed us to disentangle the contribution of relative- and phase-latency coding to odor coding. If glomerular activation times are part of the odor code, as suggested by the Hopfield model, different TOMs should elicit different odor percepts. If different percepts are indeed evoked, we should be able to assess these differences by asking the participants to describe each TOM using a set of verbal odor descriptors.

## The effect of odor temporal dynamics on odor perception

We tested participants' ability to discriminate between TOMs. First, we used two odors that are highly familiar to the majority of the population and thus can be easily named and processed as consolidated olfactory objects (*Frank et al., 2011*; *Olofsson and Gottfried, 2015*). We used extracts of orange (ORG) and cinnamon (CIN), which are naturally occurring substances often encountered in the context of food products (see 'Materials and methods'). We first tested delays of 300 ms and 600 ms between the two constituting odors, as we assumed that these delays would be long enough to generate substantially distinguishable temporal dynamics that would result in the elicitation of distinct percepts by the two TOMs. We did not test delays longer than 600 ms to ensure that the two odors would both be delivered during a single inhalation phase. Furthermore, long delays increase the likelihood that participants may realize that the TOMs are composed of two odors arriving one after the other, and this the likelihood that they may use this information for discrimination rather than the hypothesized change in odor perception itself.

We observed that out of the 10 participants who were tested with a time delay of 300 ms, nine failed to discriminate between the TOMs in a significant manner; that is, their individual performance did not differ from chance (p>0.05, binomial test per participant). Group mean and median success rates were also close to chance levels (success rate: mean = 0.541, p=0.23, t-test; median = 0.55, p=0.51, two-sided sign test, N = 10, *Figure 2A* left violet panel). An analysis based on Bayesian statistics (see 'Materials and methods') further supported this conclusion (Bayesian one sample t-test: $BF_{10}$ = 1.039, error = 8.6E-5%). When we used a longer interval of $\Delta t$ = 600 ms between odor constituents, seven out of the nine participants (78% of the cohort) failed to discriminate significantly between the TOMs. Overall, this resulted in only a slight improvement in group success rate (success rate: mean = 0.60, p=0.16, t-test; median = 0.5, p=1, two-sided sign test, N = 9, *Figure 2A*, left violet panel; Bayesian one sample t-test: $BF_{10}$ = 1.626, error = 2.3E–4%). Notably, one participant obtained a perfect score in the $\Delta t$ = 600 ms condition. In a post-session debriefing, this participant reported perceiving the two odor constituents (CIN and ORG) arriving one after the other (had we discarded this participant from the analysis, the statistical output would have changed as follows: success rate: mean = 0.55 p=0.33, t-test; median = 0.5, p=1, two-sided sign test, N = 8; Bayesian one sample t-test: $BF_{10}$ = 0.841, error = 1.9E–5%).

As it was evident that the mean was susceptible to outliers, we focused on the median statistic for the group success rate. That said, the results remained virtually the same when using the mean instead.

Participants made a similar number of wrong answers in trials in which the second TOM was the same as the first ('Same') or different ('Diff.'). This suggests that neither condition was easier than the other and that there was no strong bias in the participants' report towards one answer over the other (analysis for $\Delta t$ = 300 ms: Same = 0.505 ± 0.028, Diff = 0.495 ± 0.011, paired t-test: t(9) = 0.11, p=0.92, Cohen's d = 0.45, *Figure 2B*).

One explanation for this overall failure to discriminate between the TOMs could potentially stem from setting inappropriate parameters for the experimental paradigm (e.g., the relatively short ISI, the odor duration or the participants' motivation). To rule out these alternatives, we ran an extensive battery of control experiments using a delay of 300 ms to minimize the probability that participants would notice that the TOMs were composed of two consecutive odors. First, we verified that the

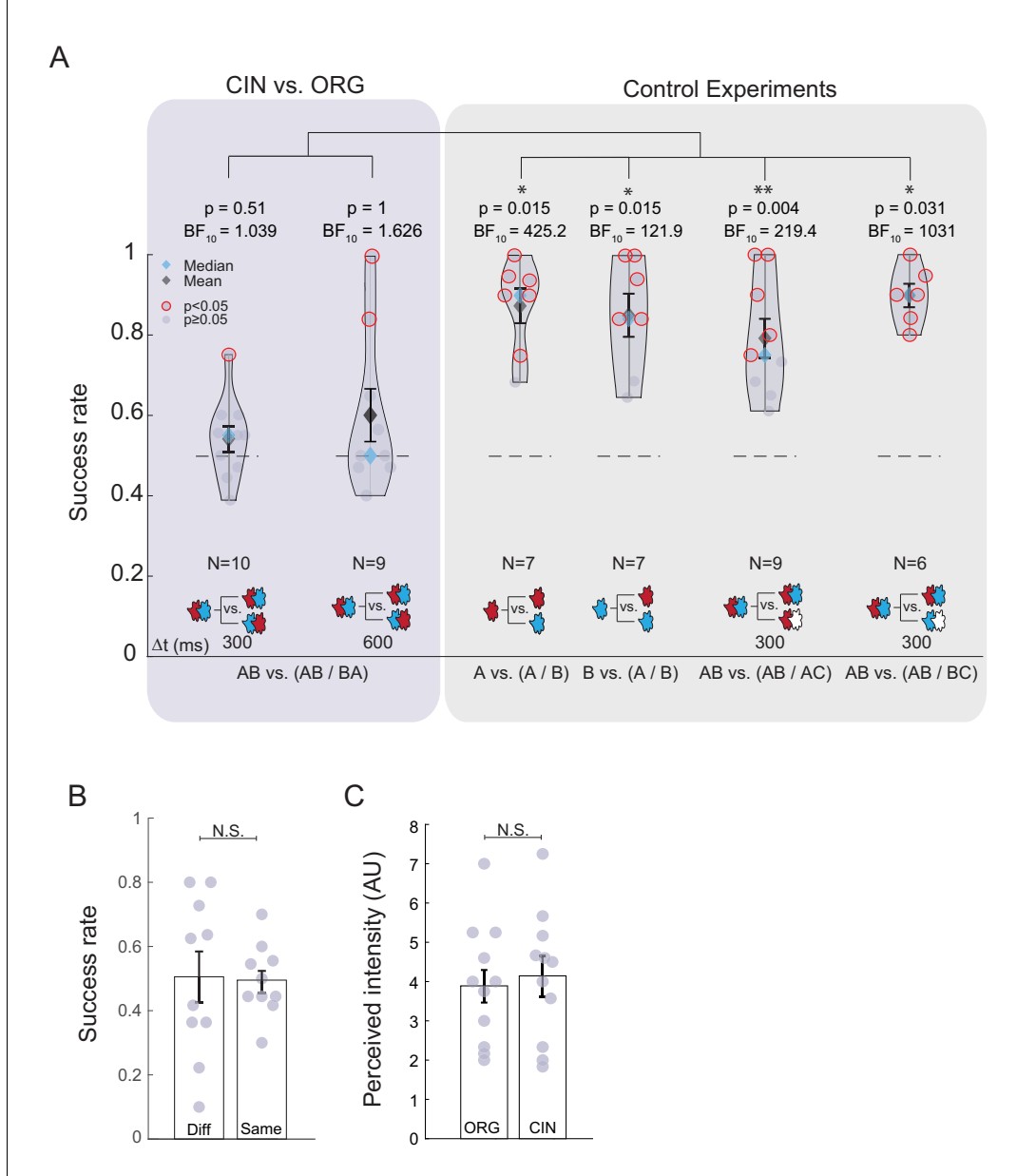

**Figure 2.** Participants could not discriminate between TOMs composed of two familiar odors. (A) Violin graph of the success rates of individual participants in discriminating between TOMs composed of ORG and CIN with Δt = 300 ms and Δt = 600 ms. Gray data points represent the success rate of an individual participant. Participants who scored significantly higher than chance are circled in red (p<0.05 binomial test, uncorrected for multiple comparisons). Means and medians are marked by black and blue diamonds, respectively. Standard deviation around the mean is marked in black. The result of a two-sided sign test, comparing the median success rate to the expected chance success rate of 0.5, and the Bayesian statistics are denoted above each group. Overlapping data points are shifted sideways across the x axis for visualization purposes. The dashed line marks the chance success rate (0.5). The number of participants in each group is denoted by N. Left violet box: participants' success rates for the TOM experiments. Right gray box: results of four control experiments validating that failure to discriminate between TOMs is not due to suboptimal choice of experimental parameters or odor delivery system. Note that in all control experiments, the majority of the cohort performed significantly above chance and the mean and median success rates were significantly above chance level. *, ** and *** represent p<0.05, p<0.01 and p<0.001, respectively (same applies for all figures in this manuscript). (B) Success rates of the TOM discrimination experiment split into trials in which the TOMs presented were the same or when they were different (Δt = 300 ms). No significant difference was detected between the two group means (two-tailed t-test, t(9) = −0.11 p=0.91). (C) Equally perceived intensity of the odors constituting the TOMs. The bar graph depicts the mean perceived intensity of the odors used (ORG and CIN). Error bars are the standard errors of the means. Gray dots denote the individual participant success rates. No significant difference was detected between the two group means (two-tailed t-test, t(10) = −0.45 p=0.66).
The online version of this article includes the following source data and figure supplement(s) for figure 2:

*Figure 2 continued on next page*

*Figure 2 continued*

**Source data 1.** Discrimination between TOMs comprised of CIN and ORG.
**Figure supplement 1.** Phase analysis.

perceived intensity of each of the constituting odors was similar. There was no significant difference in intensity (perceived intensity: ORG = 3.89 ± 0.12, CIN = 4.14 ± 0.15, paired t-test: t(10) = –0.45, p = 0.66, Cohen's d = –1.76, *Figure 2C*).

Then, we tested whether participants could discriminate between the two constituting odors when they were presented alone. We ran two experiments: 1) A vs. (A / B) and 2) B vs. (A / B). In this experiment, participants were presented with one odor and 8 s later they were presented with either the same odor or the second odor. They reported whether the two odors were the same or different. The logic behind this control was that if participants' failure to discriminate between the TOMs was due to some inadequate experimental parameter or setup accuracy, this would persist in the control paradigm, and they should also fail to discriminate between these odor sets. We found that all participants except one could discriminate between these odors (median success rate = 0.9 for A vs. (A / B). two-sided sign test p = 0.0156, Bayesian one sample t-test $BF_{10}$ = 425.2, error = undetectable, N = 7; median success rate = 0.84 for B vs. (A / B), two-sided sign test: p = 0.0156, Bayesian one sample t-test $BF_{10}$ = 121.9, error = undetectable, N = 7, *Figure 2A*, gray panel).

Finally, we ran two additional control experiments to test whether participants could discriminate between the following two odor sets: 1) AB vs. (AB / AC), and 2) AB vs. (AB / BC) (AB denotes $A \xrightarrow{\Delta t} B$, AC denotes $A \xrightarrow{\Delta t} C$, BC denotes $B \xrightarrow{\Delta t} C$, and C was a clean air stimulus of the same duration as odors A and B (200 ms)). These paradigms employed the same temporal dynamics as those in the TOMs experiment; however, here, the two odors also differed in odor content in that one TOM contained a clean air stimulus. Importantly, in these control experiments, the first odor was the same as that in the TOMs experiment (i.e., AB). In other words, this control experiment was used to validate that when the TOMs are the same, participants perceive them as same, and when they are in some way different, participants are able to notice this. The findings showed that most participants could indeed discriminate between these odors (AB vs (AB / AC) (median success rate = 0.75; two-sided sign test, p = 0.004; Bayesian one sample t-test, $BF_{10}$ = 219.4; error = undetectable; N = 9) and AB vs. (AB / BC) (median success rate = 0.9; two-sided sign test, p = 0.031; Bayesian one sample t-test, $BF_{10}$ = 1031; error = undetectable; N = 6; *Figure 2A* gray panel, right plots), although the task was slightly more difficult than discriminating between the constituting odors, probably because the two odors were more similar as they shared a component. Note that this control experiment also confirms that the constituent odors did not mask each other (at least for most of the cohort). Had this been the case, participants would have failed on the discrimination task for at least one of these odor pairs. The participants who took part in the control experiments did not participate in the TOMs experiment to prevent improvement through learning.

To compare performance on TOM discrimination and in the controls directly, we conducted an analysis of variance (ANOVA) and observed a strong main effect experiment type (F(5,42) = 10.061, partial eta-squared = 0.545, p<0.0001). Planned post-hoc comparisons (corrected with Tukey HSD) revealed that the discrimination accuracies for TOMs tested with a delay of 300 or 600 ms were similar (p=0.89) and, more importantly, that performance in TOM discrimination was significantly lower than that in all control experiments, with the exception of one comparison between Δt = 600 ms and the control AB vs. (AB/AC), which was marginally significant (p=0.067). All of the other controls maintained comparable high performance (all p-values for the TOM compared with the controls < 0.012, all p-values for cross-control comparisons > 0.68).

Note that there were fewer participants in the control experiments than in the main experiment. Nevertheless, the success rates of all four control experiments were high, whereas performance in the TOM experiments was close to chance. This indicates that the result was mostly independent of the number of participants. Moreover, the success rates in the TOM experiments were quite low (range: 50–60% for both 300 and 600 ms and for both the median and the mean, with Cohen's d = 0.43 and 0.59, respectively, *Figure 2A*). Thus, it seems unlikely that increasing the number of participants would increase mean or median success rates, although it could have made them

statistically significant. In other words, the main message conveyed by this set of experiments is that the success rates in the TOM experiments were low, whereas those in the controls were not.

Finally, the TOMs were delivered following an auditory tone, such that in each trial, the TOM could potentially be presented at a different phase of the inhalation. One potential concern is that the respiration phase in which the odor was encountered may have contributed to the odor code, so that participants may have failed to discriminate between TOMs when they were delivered at different respiration phases. That said, the fact that participants performed adequately in the control experiments suggests that phase differences were unlikely to be the reason for the failure in the TOMs experiment. We tested this directly by comparing respiratory phases across TOMs (first/second) and outcome (correct/incorrect) and observed a consistent phase distribution that was not associated with accuracy (see 'Materials and methods' and *Figure 2—figure supplement 1*).

Taken together, the results of these experiments and the ensuing analyses strongly indicate that the inability of participants to discriminate TOMs is unlikely to have been the result of an inadequate experimental parameter or setup limitation, but rather arose because the neural temporal dynamics generated in this experiment did not affect odor percept substantially.

## The effect of rapid odor temporal dynamics and odor similarity on odor perception

The results so far imply that human participants perform poorly in discrimination between TOMs when the delay between the components is 300 or 600 ms and when the constituent odors are familiar. We next tested whether the failure to discriminate between these TOMs could stem from temporal dynamics within faster time scales than the ones we used. Another possible reason for the failure in discrimination is the fact that the constituent odors, although very familiar, were both pleasant, a core property in human olfactory perception, and could therefore have been perceived similarly (*Haddad et al., 2008*; *Khan et al., 2007*), which may have rendered the TOMs hard to discriminate. In fact, 3 out of 14 participants failed to discriminate between the TOM constituting odors and 4 out of 15 failed to discriminate between a TOM and a single odor constituent (*Figure 2A*, gray panel). With this in mind, we decided to introduce a second pair of odorants, namely citral (CTL) and dimethyl-trisulfide (DMTS). These odorants have markedly more distinct percepts in that CTL is perceived as lemony/citrusy, whereas DMTS is perceived as sulfurous or onion-like (see 'Materials and methods'). In this experiment, we also tested whether shorter $\Delta t$ intervals of 150 and 75 ms along with intervals of 300 and 600 ms would have an effect, while maintaining all the other experimental parameters as in the previous experiment.

We first confirmed that this new set of odors reflected pronounced differences in percept (*Figure 3—figure supplement 1*). We then tested the participants' ability to discriminate between TOMs composed of CTL and DMTS using four delay durations ($\Delta t$ = 75, 150, 300 and 600 ms). In line with the previous experiment, the majority of the cohort (~88% or 38 out of 43 participants) failed to discriminate significantly between the two TOMs, regardless of the time delay ($p > 0.05$, binomial test per each participant, uncorrected for multiple comparisons, *Figure 3A* left box). The median for the group success rates was also not significantly higher than chance for $\Delta t$ = 75 and 150 ms and marginally significant for $\Delta t$ = 300 and 600 ms (*Figure 3A*). These results suggest that when the delay is relatively long, and the two constituent odors are dissimilar, the two TOMs can be discriminated, although the mean and median success rates were only slightly above chance and the majority of the participants failed to discriminate between them. This observation further supports the finding that temporal dynamics as elicited here do not contribute substantially to odor percept.

We next applied the same battery of control experiments used previously and observed that group performance in discrimination between the odorants was markedly higher when the odorants were not presented as TOMs. Furthermore, performance remained high for discrimination between TOMs when the second odor stimulus was replaced with clean air of the same duration ($\Delta t$ = 300 ms, *Figure 3A* right box).

A possible concern is that participants may have used the length of the first odor as a cue to facilitate discrimination in the control experiment, in which the second stimulus was replaced with air. To eliminate this eventuality, we conducted two additional control experiments. In these controls, we presented two identical pulses of the same odor constituent as the second TOM (e.g., AB vs. (AB/ AA) and AB vs. (AB/BB). In these experiments, the two TOMs had the exact same odor durations and content. Furthermore, to verify that participants did not perform well on the AB vs. (AB/AC) and

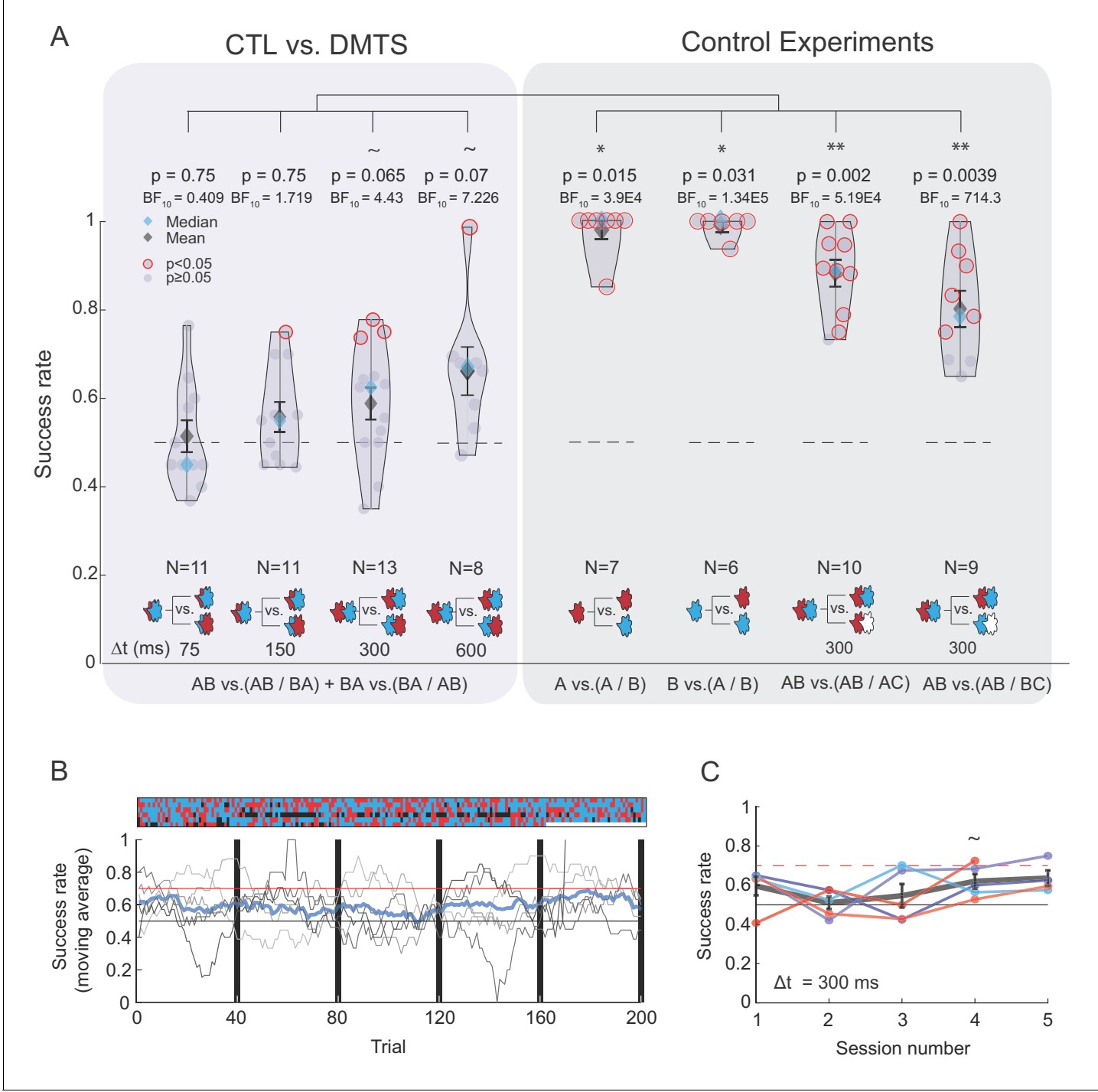

**Figure 3.** The majority of the participants could not discriminate between TOMs that are composed of highly dissimilar odors. (**A**) Left box: success rates for discrimination between TOMs composed of CTL and DMTS as a function of Δt. Color code as in *Figure 2*. Gray right box: four control experiments demonstrating that the vast majority of participants can discriminate between the constituent odors or TOMs when one of the odors is clean air (labeled as C). '~' denotes marginally significant (p>0.05 and p<0.07). (**B**) Extensive training does not substantially improve discrimination between TOMs. Lower panel: discrimination score progression over five sessions of 40 trials each. Participants who reached and maintained a score benchmark of 0.7 (red horizontal line) in one session ended their participation and they did not return for additional sessions. Chance performance (0.5) is marked by a black horizontal line. Thin gray lines mark individual participants and the thick blue line is the average of all participants over a moving window of 20 trials. The upper panel reflects trial-by-trial performance (columns) per participant (rows), where the color of each tile denotes accuracy: correct (blue), incorrect (red), or removed because the odor presentation was not in sync with sniffing (black). (**C**) Mean success rate for each participant across the five sessions. Group mean per session is depicted by a thick gray line along with vertical bars representing the standard error.
*Figure 3 continued on next page*

*Figure 3 continued*

The online version of this article includes the following source data and figure supplement(s) for figure 3:

**Source data 1.** Discrimination between TOMs comprised of CTL amd DMTS.
**Figure supplement 1.** Analysis of the percept elicited by different odor constituents used in TOMs.

AB vs. (AB/BC) control experiments by exploiting possible changes in airflow (in these experiment air was coming from a third odor port), we conducted another control experiment in which we tested whether participants could discriminate between the TOMs: AB vs. (AB/C(B)A). In this experiment, we filled the air canister (the C channel) with odorant B (denoted as C(B)) so that the TOMs were the same as those in the main experiment, but they came from different ports, as in the control experiment. We found that participants could easily discriminate between AB vs. (AB/AA) and between AB vs. (AB/BB) but did not perform above chance in discriminating between AB vs. (AB/C(B)A) (*Figure 4—figure supplement 1A–C*). Notably, in these control experiments, we used a matched-pair experimental design in which we tested the same participants in both the main experimental paradigm and the control experiment (counterbalanced for order between subjects). This further shows that the same cohort succeeded in discriminating in the control experiment but performed poorly when discriminating between the TOMs used in the main paradigm (when merged, the results were as follows: for the main paradigm, median success rate = 0.81, mean success rate = 0.82 ± 0.095; for the control experiment, median success rate = 0.55, mean success rate = 0.547 ± 0.11; paired t-test: $t(12)=8.68$, $p=1.617E–6$). Notably, one participant who performed well in both the control (100% success rate) and main experiments (83% success rate) later reported realizing that the TOMs were composed of two odors coming one after the other.

Finally, we contrasted performance in TOM discrimination across different $\Delta t$ values and controls by entering the data from all experiments into an omnibus ANOVA. There was a strong main effect for experiment type ($F(7,66) = 23.13$, partial eta-squared = 0.710, $p<0.0001$). We next carried out planned post-hoc comparisons (corrected with the Tukey HSD). These comparisons revealed that performance in TOM discrimination was significantly lower than that for all control experiments except one (AB vs. AB/BC, $\Delta t = 600$ ms, $p=0.24$). Critically, this persisted across $\Delta t$ values (mean success rate: TOM $\Delta t$ 75 ms = 0.514 ± 0.114, TOM $\Delta t$ 150 ms = 0.558 ± 0.116, TOM $\Delta t$ 300 ms = 0.588 ± 0.13, TOM $\Delta t$ 600 ms = 0.666 ± 0.158, A vs. A/B = 0.978 ± 0.052, B vs. A/B = 0.948 ± 0.091, AB vs. AB/AC = 0.883 ± 0.092, AB vs. AB/BC = 0.802 ± 0.115.; all other p values of TOM compared with controls < 0.0042). To recapitulate, our results so far portrayed a picture in which participants did not perceive TOMs to be different enough for easy discrimination.

## Extensive training does not substantially improve discrimination between TOMs

Previous experiments in rodent models reported that the exploitation of temporal features typically involved extensive training of the animals, sometimes for hundreds of trials, while maintaining high motivation (for example, through water deprivation). Given this experimental regimen, it is worth inquiring whether after extensive training, or when motivation is high enough, more olfactory information is extracted from odor-elicited temporal features that contributes to task performance. To probe this possibility, we replicated the same experimental paradigm in a new cohort of five participants, but this time, we asked them to take part in a total number of up to 200 trials (two sessions per day, 40 trials each, over three days). We used the same two highly dissimilar odors as before (CTL/DMTS) and, as before, there was a fixed delay of $\Delta t = 300$ ms.

Bearing in mind the obvious differences in reward circuitry between rodents and undergrad students, we attempted to increase participants' motivation by offering a monetary bonus (e.g., doubling the session payment) upon hitting a performance benchmark of clearly succeeding in discriminating between the two TOMs; that is, reaching and maintaining a success rate of 0.7 during a 40-trial session. Participants who reached this goal were fully rewarded and did not have to return for additional sessions. Mean success rates per day ranged from 0.51 to 0.64 and were not significantly different from chance. Critically, they did not improve as a function of session (two-sided sign test $p>0.05$ in 4 out of 5 sessions; note the marginal significance of cohort performance in the fourth

session, p=0.0625, N = 4) (*Figure 3B–C*). This result suggests that the percepts evoked by the two TOMs were still mostly indistinguishable, even after extensive training.

## The odor of TOMs is perceived as an intermediate of its components

Our behavioral results indicate that TOMs seem to be perceived as similar, even when the delay between the constituent odors is long (e.g., $\Delta t$ = 300 or 600 ms) making it hard to discriminate between TOM AB and TOM BA. To assess odor perception elicited by the TOMs directly, we asked a new group of participants (N = 21) to rate the TOMs and their constituent odors on a list of eleven verbal odor descriptors that are commonly used in olfactory psychophysics (*Dravnieks, 1982*) (see 'Materials and methods'). TOMs were delivered using the same methodology as before, at a time delay of $\Delta t$ = 300 ms (N = 12) or $\Delta t$ = 600 ms (N = 9), with participants able to undergo more than a single smelling round before rating each TOM. To eliminate possible habituation, the time delay between smelling the two TOMs was set to at least 23 s.

We next projected the odor ratings for the two TOMs and their constituting odors onto a two-dimensional space using principal components analysis (PCA). We observed that the two TOMs were perceived as an intermediate percept, positioned in space between the constituting two odor perceptions (*Figure 4A*). Calculating the Euclidean distance between the average descriptor ratings of the four odor stimuli (e.g., A, B, AB, and BA for $\Delta t$ = 300 and 600 ms) confirmed that while the two constituting odors were markedly different, the perceptual distance between the two TOMs was much smaller ($\Delta t$ = 300 ms: distance (AU) A vs. B = 14.24 ± 4.09, AB vs. BA = 10.11 ± 3.48; two sample t-test, t(286) = −9.21; Cohen's d = 1.08, p=7.35E–18; *Figure 4B*). An analysis of distance hierarchy between all stimuli confirmed the existence of a significant main effect for odor type (F(5,858) = 23.25, p<0.0001). Post-hoc comparisons confirmed that the perceptual distance between the mono-molecules was larger than for all other stimuli (all p<0.0001, Tukey HSD corrected) (*Figure 4C*). In other words, whenever a similarity rating included a TOM, it was not significantly different from other similarity ratings that included a TOM, and the perceptual distance between the mono-molecular constituents stood out in all of these comparisons. In addition, the perceptual distance between the two TOMs was smaller than all other compared distances except one (all p<0.05, Tukey HSD corrected, with the exception of AB vs. BA and A vs. AB, p=0.61). When the TOMs were delivered at a delay of 600 ms, a similar picture emerged in which the perceptual distance between the two TOMs remained distinctly smaller than the distance between their mono-molecular constituents (*Figure 4D*) (for $\Delta t$ = 600 ms: Distance (AU) A vs. B = 18.86 ± 4.33, AB vs. BA = 10.98 ± 3.39; two sample t-test, t(160) = −7.98; Cohen's d = 1.25, p=2.61E–13; *Figure 4E*). An analysis of distance hierarchy between all stimuli confirmed the existence of a significant main effect for odor type (F(5,480) = 18.31, p<0.0001). Post-hoc comparisons confirmed that even with a 600 ms interval between components, the perceptual distance between the mono-molecules was larger than that for all other stimuli (all p<0.0001, Tukey HSD corrected) (*Figure 4F*).

## Odor-elicited temporal dynamics can be used to extract odor-related information

We next examined whether participants could use explicit information regarding the temporal structure of the stimuli to facilitate discrimination. We therefore informed the participants prior to the session that the two TOMs were composed of two odors delivered one after the other in a different order. We examined two variations: in the experiment composed of the similar, more nameable odors, participants were told that the TOMs were composed of cinnamon (CIN) and orange (ORG). In the experiment presenting dissimilar odors (CTL and DMTS), participants were told that the one odor was pleasant and the other less so.

To test whether awareness of the TOM temporal composition affected performance as a function of $\Delta t$, we conducted an ANOVA separately for each odorant pair (i.e., ORG/CIN or CTL/DMTS) with the variable '$\Delta t$' (75/150/300/600 ms). For ORG/CIN, we observed a significant main effect for '$\Delta t$', implying that the success rate varied considerably as a function of the TOM $\Delta t$ (F(3,44) = 6.05, partial eta-squared = 0.292, p=0.0015, N = 48, *Figure 5A*). A parallel analysis conducted on the CTL/DMTS odorant pair indicated a similar significant main effect for '$\Delta t$' (F(3,28) = 19.94, partial eta-squared = 0.681, p<0.00001, N = 36, *Figure 5B*).

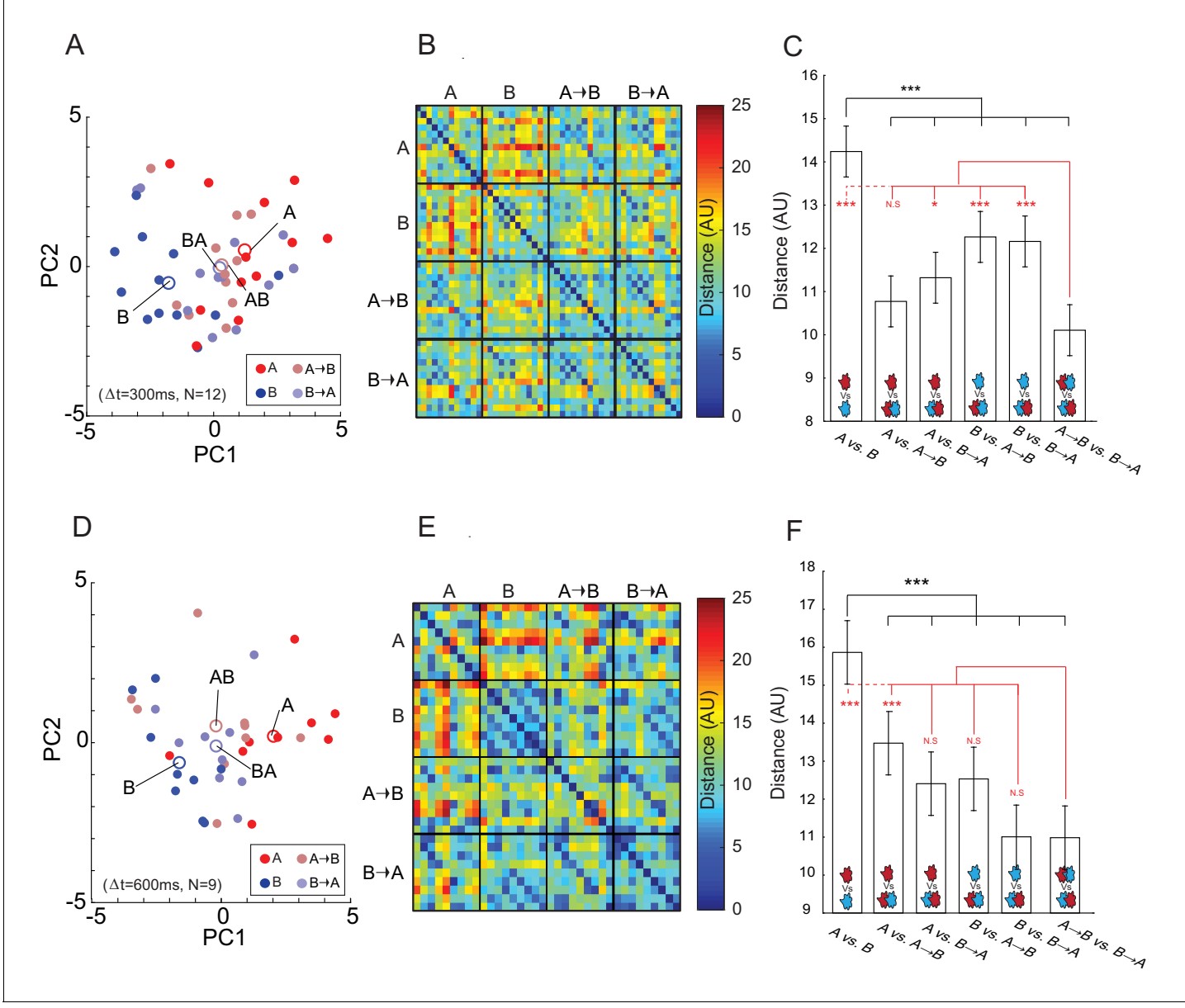

**Figure 4.** Perception of TOMs is an intermediate of that of their components. (A) Projection of verbal descriptor ratings of the four odors onto the two main axes of principal component space for CTL (labeled 'A', red circles), DMTS (labeled 'B', blue circles) and two TOMs comprised of CTL and DMTS presented in two temporal sequences: $A \underset{\Delta t=300}{\rightarrow} B$ (brown) and $B \underset{\Delta t=300}{\rightarrow} A$ (purple). Each data point represents the ratings of a single participant for a given odor or TOM. The centroids of each cluster are marked by labeled circles following the same color scheme. (B) A matrix comprised of the Euclidean distances between eleven descriptor ratings provided by the participants for CTL (labeled 'A'), DMTS (labeled 'B') and two TOMs comprised of CTL and DMTS. Hotter colors denote larger distances. Ratings are divided into four odorant subgroups by a black grid. Mosaic-like patterns within each compartment represent the between-participant variability of ratings for the same odor. (C) TOMs are perceived more similarly to each other than to their isolated constituents. Bar graph of mean perceptual distance between TOMs ($A \underset{\Delta t}{\rightarrow} B$ or $B \underset{\Delta t}{\rightarrow} A$ ) and their constituent odorants (A and B).

Statistical significance is denoted for post-hoc comparisons of perceptual distance between the monomolecular constituents ('A vs. B') with all other stimuli in black and for the TOMs with all other stimuli in red. Error bars are mean ± SEM. (D–F) Same as panels (A–C) for Δt = 600 ms.
The online version of this article includes the following source data and figure supplement(s) for figure 4:

**Source data 1.** Olfactory verbal descriptors for TOM perception.
**Figure supplement 1.** Additional control experiments.

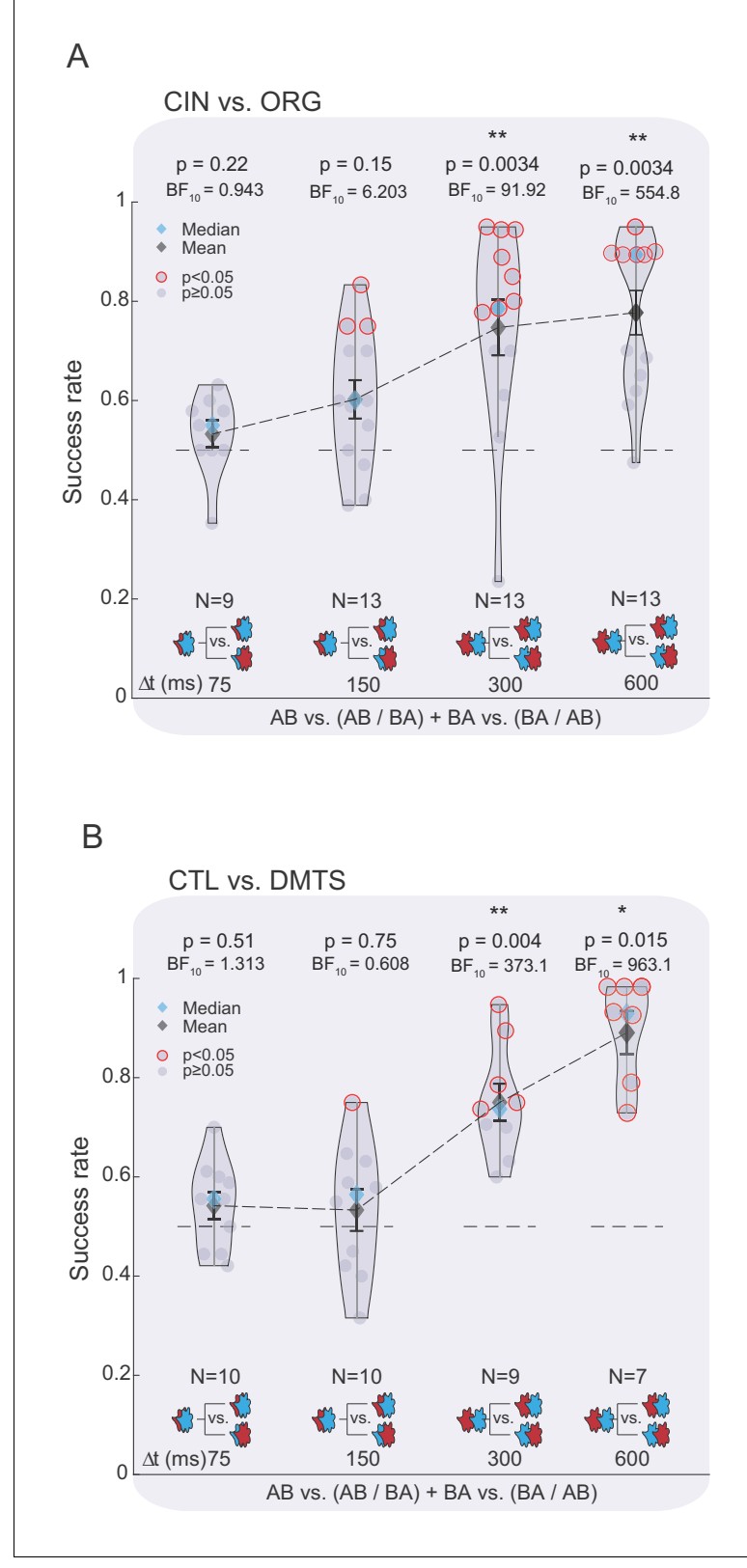

**Figure 5.** Temporal odor dynamics can be used to extract odor-related information. (**A**) Success rate of discrimination between the TOMs constituting the ORG and CIN odors as a function of Δt when participants were aware of the constituent odors and their temporal features. Color and legend code as in *Figure 2*. (**B**) Same as in panel (**A**), but for the TOMs composed of CTL and DMTS.

*Figure 5 continued on next page*

*Figure 5 continued*

The online version of this article includes the following source data for figure 5:

**Source data 1.** Discrimination between TOMs when aware of temporal features.

In other words, when participants were explicitly instructed that the incoming odor stimuli were composed of two consecutive odorants, we observed an incremental improvement in both individual participants' success rates and the group's median performance. This trend was similar for both odor pairs despite minor variations (in the Δt = 150 ms delay, performance was better in the ORG/CIN odor set than in the CTL/DMTS, yet the Δt = 600 ms exhibited the opposite trend; *Figure 5A–B*). This 'unlocking' of discrimination ability suggests that temporal dynamics can be utilized to extract odor information.

Thus, overall, glomeruli activation times do not elicit a prominent difference in odor percept; however, they can be used to extract temporal odor information such as detecting the order of odor delivery in a sequence.

## Discussion

### Effects of latency coding on odor perception in humans are inconsequential

Odor-elicited temporal dynamics that are not directly related to stimulus dynamics have long been observed in the olfactory system. These observations gave rise to the hypothesis that odors are encoded by a spatiotemporal code. What a temporal code means in the context of the olfactory system has been interpreted in several ways; one prominent model suggests that the activated glomerulus time relative to some internal or external event is part of the odor code. In the current study, we tested whether the time of glomeruli activation affected odor perception as predicted by this model. We used odor stimuli composed of two odors sequentially presented in different orders (TOMs) and observed that most of the participants had a poor ability to discriminate between TOMs (*Figures 2A*, *3A and B*).

To mitigate any concerns that the rather poor performance in the task was a result of some inadequate experimental condition, we employed a comprehensive battery of control experiments (*Figures 2A* and *3A* gray panels, and *Figure 4—figure supplement 1*). These experiments showed that participants performed well when the two odors differed not just in terms of their temporal order. Note that the success rates in the control experiments were high, despite the small cohort, compared to the main experiment, suggesting that the number of participants was adequate to allow this conclusion to be drawn. In addition, performance rates improved dramatically when the participants became aware of the constituent odors (*Figure 5*), further indicating that the failure to discriminate TOMs was not due to technical shortcomings.

We chose to conduct these experiments with human participants to take advantage of a robust and widely accepted supposition that humans can tell what they smell. In fact, an analysis of the percept elicited by each TOM and their constituent odors confirmed that temporal mixtures evoked odor percepts that were in between those of the two constituent odors, regardless of their order of presentation (*Figure 4*). This type of experiment is significantly more challenging to interpret in animal models, because even when performance is adequate, one cannot attribute this to the different percepts evoked by the TOMs or to an ability of the animals to identify the TOM temporal structure, that is, to identify that two odors have been presented consequently. Importantly, even if under some conditions participants could have discriminated between TOMs, the fact that the TOMs were perceived as similar rather than as generating a new odor percept is strong evidence that temporal coding (of the type examined here) is not a substantial part of the odor code per se.

### Temporal coding models

We view these results as more consistent with one possible interpretation of the Laurent model — that time is not necessarily part of the code — and less so with the Hopfield model that time is part of the code. Moreover, the results do not align with a temporal coding model in which the first activated glomeruli form the odor code (*Schaefer and Margrie, 2007*; *Wilson et al., 2017*). We argue

that if that were the case, participants should have easily discriminated between the TOMs, as the first-activated glomeruli in the TOM $A\underset{\Delta t}{\longrightarrow}B$ were most probably very different from those activated in the TOM $B\underset{\Delta t}{\longrightarrow}A$.

One possible interpretation of the Hopfield model that can be reconciled with our results is that although the time of glomeruli activation is not part of the odor code, the time of glomeruli activation in relation to the internal gamma oscillation or the exact spike times of each neuron are indeed part of the code. The odor manipulations in this study could not have affected spike time relative to the gamma oscillation cycle, so this remains a viable possibility.

## Temporal dynamics can be used to exploit odor-related features

When participants became aware of the temporal structure of the TOMs and their constituent odors (i.e., rapid onset of two consecutive odors) before the session, we observed higher performance in discrimination when the Δt was set to 300 and 600 ms, but the participants still performed around chance when this delay was shortened to 75 ms to allow for a more rapid succession (*Figure 5*). This may suggest that although the latency code only weakly affects the odor percept of the TOM, it could be used to disentangle the constituent odors, provided that the sequential nature of the stimuli is disclosed, and the temporal dynamics do not evolve too rapidly. One possible interpretation of this result is that when a delay was introduced, there was a substantial duration of time in which there was no odor at all. This pause between the two odors might have been used to detect the existence of two odors and might therefore have contributed to perceiving them as two odors delivered one after the other. Another possible explanation is that participants performed well because they employed a pattern-matching algorithm. A few participants reported that they were actively searching for a specific odor to occur at the beginning or end of the stimulus, suggesting that they employed a matching algorithm for one of the constituents. When the delay was set to be shorter than 150 ms, this matching failed because the first odor was presented in a partial temporal overlap with the second one. This strategy is also in line with a previous study in which participants were able to name which odor out of two known odors was presented first when the delay was 200–400 ms (*Laing et al., 1994*) or when it was presented in the presence of a background stimulus (*Smith et al., 2010*). The mechanism governing this is currently unknown, but we speculate that top-down mechanisms that either shut down background activity or improve sensitivity for the target odor are likely to be involved. These results thus suggest that glomeruli activation time can be utilized to extract odor-related information such as when the odor was delivered relative to other odors or relative to respiration phase.

## Human versus non-human organisms

Our experiments were conducted in humans who are generally (and some say wrongfully) regarded as a microsmatic species (*Rouquier et al., 2000*). One can speculate that in mascrosmatic animals who rely more heavily on their sense of smell, such as rodents and insects, temporal coding is part of the odor code. Several studies have recently probed the ability of mice to exploit temporal coding to extract odor information. Owing to the considerable challenge posed by delivering two odors within the short inhalation time of mice (typically <100 ms in actively smelling mice), the authors conducted these experiments using optogenetic stimulation in mice expressing Channelrhodopsin2 in the olfactory sensory neurons (OSN) or the M/T cells. Mice learned to discriminate between two light stimulations of the OSNs, M/Ts or even a single glomerulus that were a few milliseconds apart, either relative to, or irrespective of respiration (*Rebello et al., 2014*; *Smear et al., 2011*; *Smear et al., 2013*). These experiments clearly demonstrated the use of timing in the olfactory system of mice. However, they did not demonstrate that mice perceived the optogenetic stimulations as two different odor percepts as mice cannot report their odor perception directly. One possible interpretation is that the trained mice perceived the two optogenetic stimuli as the same odor presented at different respiration phases. Alternatively, the mice may have perceived the optogenetic stimuli as two different 'odor' durations or two odors delivered one after the other. Furthermore, the behavioral results from non-human model systems frequently rely on extensive training (e.g., thousands of trials performed by highly motivated animals), thus making it hard to distinguish between what the olfactory system does and what the olfactory system can do. It is thus possible that although latency coding does not substantially affect odor perception, temporal dynamics can

be utilized under extreme conditions. This may be achieved through neurons that are sensitive to differences in M/T-relative activation times (*Haddad et al., 2013*).

## Limitations

Our study is not without limitations. First, for practical reasons, we focused on a limited set of odor mixtures that were always composed of only two odorants. Second, by using TOMs, we could not change the spike timing of individual olfactory neurons. Several studies have shown that a substantial amount of odor information can be extracted from 'sub-sniff' neuron spike timings (*Bathellier et al., 2008*; *Bolding and Franks, 2017*; *Cury and Uchida, 2010*; *Sirotin et al., 2015*). It thus remains a clear possibility that the odor-elicited spike timing of each neuron is a carrier of odor identity. This coding scheme suggests that changing the spike timing of a particular neuron or neurons will result in a different odor percept. This opens up exciting new possibilities, but it remains a daunting technical challenge. Finally, it is still possible that the participants failed to discriminate between TOMs as a result of some technical shortcoming in our odor-delivery system or some other caveat that we were unaware of. This is always a possibility in experimental research, but we find it unlikely as participants discriminated between the constituent odors with ease (*Figures 2B* and *3*), and were able to discriminate between TOMs when they were made aware of their temporal composition (*Figure 5*) or when we replaced one odor with the presentation of clean air or the same odor again (*Figures 2A* and *3A* and *Figure 4—figure supplement 1*).

Thus, overall, whether the temporal dynamics observed in odor-elicited neural responses in rodents are part of the odor code remains an ongoing debate. Our experiments in humans indicate that latency coding could be utilized to extract odor-related information without substantially affecting odor perception.

# Materials and methods

## Participants

This manuscript describes a series of experiments conducted on an overall cohort of 284 participants (191 female, 93 male, age range 18–41 years). All participants were healthy, self-reported to be normosmic and in good health at the time of the experiment. None of the female participants were pregnant at the time of the experiment (self-report). Participants were all university students, recruited via advertisement on campus grounds. Written informed consent and consent to publish was obtained from the participants in accordance with the ethical standards of the Declaration of Helsinki (1964). The experiment was approved by the institutional ethics committee of Bar Ilan University (reference number ISU20140804001). Participants were paid for their participation. To avoid cross-learning, each participant was tested using only one condition with exception of the experiments reported in *Figure 4—figure supplement 1*.

## Experimental design

The temporal odor mixtures (TOMs) were comprised of fragrant oil mixtures of orange (ORG) and cinnamon (CIN), or of the monomolecular odorants Citral (CTL) (CAS: 5392-40-5, Sigma-Aldrich) and Dimethyl Trisulfide (DMTS) (CAS: 3658-80-8, Sigma Aldrich). In both cases, odors were diluted to equally perceived intensity (CTL: 1:100, ORG: 1:1, DMTS, 1:10000, CIN: 1:1). The compositions of the fragrant oil mixtures were analyzed using gas-chromatography mass-spectrometry (GCMS). Tenax tubes were used for trapping volatiles, and these were subjected to GCMS combined with dynamic headspace sampling. The main constituents of each oil according to their relative contribution to the mixture were as follows. Cinnamon oil: E-cinnamaldehyde (CAS 14371-10-9), linalool (CAS 78-70-6), o-cymene (CAS 527-84-4), terpineol (CAS 98-55-5) and cinnamyl acetate (CAS 103-54-8). Orange oil: limonene (CAS 138-86-3), gardenol (CAS 93-92-5) and linalool (CAS 78-70-9).

We used a computer-controlled air-dilution custom-built olfactometer to deliver odors in varying orders and sequence lengths. Odor stimuli lasted 200 ms. The inter-stimulus interval (i.e., the time between two TOMs) was set to 8 s. The next trial (i.e., the next pair of TOMs) was presented 23 s after the participant submitted his or her answer (which usually lasted ~2–5 s). Three short tones (1 s) informed the participant about the incoming odor presentation. Participants underwent several training trials to practice the synchronization of a nasal inhalation longer than 1 s, just before the odors

were delivered. To simplify the task, participants were instructed that the first TOM was fixed. Participants were allowed three trials to familiarize themselves with the two TOMs and the experimental setup, and then conducted 20 trials. Nasal respiration was recorded using a nasal cannula attached to a pressure-sensitive transducer, which translated these changes in pressure into an electrical signal via a USB interface (*Plotkin et al., 2010*). The signal was acquired and digitized at a sampling rate of 1 KHz. The recording of respiration and odor triggers and the user interface were designed and carried out in a LabView environment (National Instruments).

## Odor delivery

Odor and air stimuli were delivered via a three-port custom-built odor-delivery system. To prevent cross effects between the two odor ports (e.g., air pressure changes or odor contamination), we used three completely independent channels (*Figure 1A*). To deliver an odor, purified air (medical grade 99.999%, Maxima, Israel) was streamed through a glass vial containing a liquid odor. Odor selection was done using a digitally controlled solenoid valve (07V113, AIGNEP) on each of the dedicated delivery channels. One-way check valves were used to prevent backflow of the odorized air stream. The air flow was set to 1.8 LPM using a mass flow controller (Alicat Scientific). Odors and the odor-delivery plastic tubing were replaced every day.

To prevent contamination, participants were seated in front of three dedicated odor ports, one for each of the odorants (the distance from the odor ports was ~5 cm, *Figure 1A*). We used a relatively short delay (8 s) between the first and the second TOM because pretesting showed that longer delays between the two TOM presentations diminished the success rates, even when the participants were asked to discriminate between two very different odors. We verified that this short ISI was adequate by testing participants' ability to discriminate between the two pairs of odors (*Figure 2*). In the first set of experiments (ORG and CIN), the first TOM was set to be AB and the second TOM was either AB or BA. In part of the second set of experiments (CTL and DMTS), we changed the first TOM to BA. We modified this design to make sure there that was no bias related to the identity of the first TOM. As the results were the same, we pooled the data.

To ensure that odor presentation matched the inhalation phase of the respiratory cycle, we instructed participants to inhale for a duration of at least 1 s just before the odor was presented (using an auditory cue prior to each odor arrival). Trials in which the odor presentation extended into the exhalation period or started prematurely before inhalation were excluded from the analysis (*Figure 1F*, methods). The delay between the two odors, Δt, was set to ≤600 ms. We did not test longer delays because the overall duration of a TOM had to be shorter than a typical inhalation period (about 1–1.5 s) to be perceived in full and to minimize the possibility of discriminating between the two TOMs by realizing that they were composed of two consecutive odors. In fact, at long delays (e.g., 300 and 600 ms) a few participants realized that the TOMs were composed of two consecutive odors. We did not exclude these participants from our analysis to prevent bias.

## PID measurements

To estimate odor concentration, we used a mini photo-ionizer detector (miniPID, Aurora Scientific). PID response amplitude depends on the odor identity, its concentration and the distance from the inlet. Each odor elicits a different response that is related to how effectively the measured odor can be ionized. We measured the door PID responses to the odors and their temporal dynamics at different delays. All TOMs elicited similar response dynamics. PID responses varied between these delays and the order of odor presentation, reflecting the between-odor interactions and inter-pipe flow interactions. However, the odor concentration of the odor constituents and the TOMs in three tested delays was highly similar across trials of the same condition. Thus, each TOM was expected to elicit the same odor percept across all trials. Measurements of odor concentration when placing the PID inlet ~5 cm from the odor ports (i.e., the location of the participants' nose) revealed that the order of odor presentation was preserved, although the odor concentration was more variable across trials when measured at a 5 cm distance (*Figure 1—figure supplement 1C*).

## Post-processing and analysis

The respiration signal was zeroed and baselined such that positive values denoted inhalation and negative values denoted exhalation. A 0.05–100 Hz band pass filter was applied to the respiratory

trace to remove high-frequency artifacts and drift. Next, following normalization by a z-score, the entire recorded trace underwent segmentation into epochs, each consisting of a single trial.

To be sure that the odor pulses arrived in synchrony with the inhalation period, we meticulously verified all odor pulses occurring within the inhalation period. We scored the proportional duration of the odor pulses within each TOM, or in other words, how much of the odor pulse was presented concurrently with nasal inhalation. An automatic algorithm assigned a score to each trial, according to the combined score of all odor triggers in a given trial. We applied a strict criterion that set the cut-off to 1.00 (all odor pulses were presented fully within inhalations) to reject any trials that were questionable. Finally, this automated process was backed by a trial-by-trial visual inspection. The results remained the same when we used a less stringent cut-off (e.g., 0.9) or when we assumed that odor offset was longer than the estimate obtained from the PID measurements (e.g., assuming a very long odor offset of 200 ms).

In the main experimental paradigm, each session consisted of 20 trials. Any participant with a total number of fewer than 14 valid trials (70% of the session) was excluded from further analysis. This led to the exclusion of 73 participants following analysis of their trial-by-trial data, leaving a total of 284 participants. The number of eliminated trials within the pool of remaining participants who took part in the TOM paradigms totaled 162 out of 1940 and 223 out of 3972 trials, or 8.3% and 5.6% of all events in the CIN/ORG and CTL/DMTS experiments, respectively. The results were virtually the same when we used a less stringent criterion for excluding trials.

As detailed above, two TOMs were presented in each trial. Each was comprised of two rapid odor pulses spaced apart by a pre-defined delay. Odor stimuli were presented subsequent to an auditory cue instructing the participants to inhale. The stimuli were not, however, triggered to lock with a certain phase of nasal inhalation, and as a result, were encountered at different respiration phases. Respiratory phase was calculated with MATLAB's 'angle' function applied to the Hilbert transform of the respiratory trace. The product of this calculation is the phase that gradually increases from $-\pi/2$ to $\pi/2$ over the course of the inhalatory phase of the respiratory cycle (see *Figure 2D* for visualization).

## Statistical analysis

Given the two-alternative nature of the task, the outcome of each trial could either be 'correct' or 'incorrect', and the success rate at chance level was 0.50. A success rate significantly higher than chance in a 20-trial experiment was therefore calculated to be 0.70 (binomial cumulative distribution function). In cases where the number of valid trials was lower than 20, this threshold was adjusted accordingly, such that it significantly exceeded chance at a significance level of $p < 0.05$. Similarly, the median group performance was compared to chance (0.50) using a two-sided sign-test. A comparison of group performance across conditions was carried out using an analysis of variance (ANOVA). To reduce the effect of outliers on the result of this study, we focused our analysis on the group medians. However, the results remained the same even when we used the group mean (i.e., performed a one-sample two-sided t-test).

In addition to standard testing of the data against a null hypothesis, we also subjected the data in each analysis to Bayesian one-sample t-tests with success rate as the dependent variable, compared to chance performance (0.50) with a Cauchy prior of 0.707 (*Good, 1962*) The added insight gained from this approach stems from its ability to quantify the evidence in favor of two different models. Bayesian statistics are advantageous in assessing the relative probability of the null hypothesis over the experimental hypothesis. This advantage becomes a necessity when one does not reject $H_0$ (i.e., 'non-significant results') and needs to quantify the evidence to support this claim (*Leech and Morgan, 2002*). We therefore detailed our Bayesian statistics alongside each regular sign-test. The output Bayesian statistic used was the $BF_{10}$, which depicts an odds ratio; namely, the probability, or simply how likely the data are under both hypotheses. In our interpretation, we used the standard recommendation that a $BF_{10}$ between 1 and 3 implies anecdotal evidence, 3–10 substantial, and 10–30 strong evidence, where $BF_{10}$ quantifies evidence for the alternative hypothesis relative to the null hypothesis. All the Bayesian statistical analyses were conducted in JASP (2019) version 0.9.2. Statistical analyses concerning the values of the respiratory phase were carried out using functions implemented in CircStat MATLAB, a toolbox for circular statistics that are analogous to the regular t-test or ANOVA (*Berens, 2009*).

## Odor similarity analysis

To estimate olfactory perceptual distance between all odors, we asked a separate cohort (N = 12 for $\Delta T$ = 300 ms and N = 9 for $\Delta T$ = 600 ms) to rate all four odors used in this study (ORG, CIN, CTL and DMTS) on the basis of eleven verbal descriptors curated from a list of commonly used descriptors (*Dravnieks, 1985*). To calculate odor similarity, we then projected participants' ratings into a two-dimensional space using principal component analysis (PCA), a dimensionality reduction method common in olfactory research (*Haddad et al., 2010*; *Khan et al., 2007*). As expected, ORG and CIN odors encompassed an overlapping area in this perceptual space, whereas CTL and DMTS were markedly divergent, with CTL more similar to ORG/CIN (*Figure 2—figure supplement 1A*). We quantified similarity by calculating the mean Euclidean distance between the ratings for each odor. The perceptual distance between CTL and DMTS was significantly higher than the distance between ORG and CIN (CTL/DMTS = 5.52 ± 1.12; CIN/ORG = 4.18 ± 1.25, paired *t* test: t(110) = 5.95, p=3.1E–8, Cohen's d = 1.12, *Figure 2—figure supplement 1C*). Furthermore, this perceptual distance was supported by a predictive algorithm allowing for the estimation of perceptual similarity from molecular structure (*Snitz et al., 2013*). The distance between ORG and CIN was 0.0189 radians, but the distance between CIT and DMTS was 1.0846 radians. In other words, discriminating between TOMs composed of citral and DMTS was expected to be an order of magnitude easier than discriminating between ORG and CIN. Last, as with the previous TOMs, we verified that these two odors had similar intensities (CTL = 7.37 ± 1.19, DMTS = 7.12 ± 0.99, paired t-test, t(7) = −0.84, Cohen's d = 0.23, p=0.43, N = 8).

## TOM phase analysis

We computed the respiration phases of the two TOMs for correct and incorrect trials. The respiration phases for the first and second TOMs were correlated, both for correct and incorrect trials (correct: circular correlation r = 0.67, p=1.33E-8, N = 100; incorrect: r = 0.549, p=1.02E-5, N = 84). The regression line slopes were close to one (correct: a = 0.942, incorrect: a = 0.859), indicating that participants tended to encounter the two TOMs at similar respiration phases (*Figure 2—figure supplement 1A–B*). Moreover, comparing the differences between the TOMs' respiration phases (e.g., TOM1 – TOM2) in correct and incorrect trials further showed that the phase differences in the correct and incorrect trials were not significantly different (non-parametric multi-sample test for equal medians, Kruskal-Wallis test for circular data: shared population median = −0.478 rad, KW(P)=1.402, p=0.236; *Figure 2—figure supplement 1C–D*). Finally, an adaptation of the Kolmogorov-Smirnoff test for circular data (Kuiper test) conducted iteratively 10,000 times on the phase data suggested an average p-value of 0.985, with an average k statistic of 480.15 ± 126.4; in other words, the phase distributions were highly similar.

## Olfactory perception of TOMs

Odorant and TOMs were rated using a set of eleven verbal descriptors (pleasant, fruity, edible, hot, chemical, medicinal, smoky, alcoholic, attractive, earthy, and sulfurous) on a scale of ranging from 0 to 10 (where 0 corresponded to 'not at all' and 10 corresponded to 'very much'). It should be noted that in rating sessions involving TOMs, these were always rated before the monomolecular odors in order to prevent bias in the perception of the mixtures, given that their isolated components had not yet been presented separately. Participants could undergo several smelling rounds of the same odor before rating, and an inter-stimulus interval of 23 s was imposed between any two odor presentations. Data for two descriptors of a single participant were corrupted and were replaced by the group average for that score. Prior to projection into the principal component space, descriptor ratings were normalized using the Z-score.

## Acknowledgements

We would like to extend our thanks to Shani Agron for help with GCMS analyses. We are indebted to Noam Sobel for helpful insights provided during the preparation of this manuscript. This study was supported by the ISF- 204/17 grant.

## Additional information

### Funding

| Funder | Grant reference number | Author |
|---|---|---|
| Israel Science Foundation | 204/17 | Rafi Haddad |

The funders had no role in study design, data collection and interpretation, or the decision to submit the work for publication.

### Author contributions

Ofer Perl, Conceptualization, Data curation, Software, Formal analysis, Investigation, Visualization, Methodology; Nahum Nahum, Conceptualization, Data curation, Software, Formal analysis; Katya Belelovsky, Data curation, Project administration; Rafi Haddad, Conceptualization, Formal analysis, Supervision, Funding acquisition, Validation, Investigation, Visualization, Methodology

### Author ORCIDs

Ofer Perl (iD) https://orcid.org/0000-0002-3560-4344
Katya Belelovsky (iD) https://orcid.org/0000-0001-9960-7336
Rafi Haddad (iD) https://orcid.org/0000-0001-8285-5210

### Ethics

Human subjects: Participants were all university students, recruited via an advertisement on campus grounds. Written informed consent and consent to publish were obtained from participants in accordance with the ethical standards of the Declaration of Helsinki (1964). The experiment was approved by the institutional ethics committee of Bar Ilan University (reference number: ISU20140804001). All experimental sessions were conducted after obtaining informed consent and the participants were paid for their participation.

### Decision letter and Author response

Decision letter https://doi.org/10.7554/eLife.49734.sa1
Author response https://doi.org/10.7554/eLife.49734.sa2

## Additional files

### Supplementary files

• Transparent reporting form

### Data availability

Source data for Figures 2–5 have been provided.

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
