## [Decision Letter]

**Acceptance summary:**

This paper studies whether humans use time-differences in odor arrival as part of the precept of odor discrimination. The study makes the surprising case that odor arrival time does not contribute strongly to odor identification, and this finding runs counter to considerable prior work albeit mostly in rodents. Because this claim is surprising, and the readouts are behavioural and lack access to physiology in human subjects, the report has been supported with a comprehensive set of controls. These reveal the suggestive nuance that when humans are primed to look for sequential odor delivery, their performance on the exact same task improves. It will be interesting to see how this set of findings is reconciled with the rodent literature.

**Decision letter after peer review:**

Thank you for submitting your article "The contribution of temporal coding to odor coding and odor perception in humans" for consideration by *eLife*. Your article has been reviewed by three peer reviewers, one of whom is a member of our Board of Reviewing Editors, and the evaluation has been overseen by Catherine Dulac as the Senior Editor. The reviewers have opted to remain anonymous.

The reviewers have discussed the reviews with one another and the Reviewing Editor has drafted this decision to help you prepare a revised submission.

The reviewers all agreed that the results were surprising and potentially interesting. They brought up a number of technical and conceptual points that they felt could be cleared up by some rather straightforward experiments of the same design.

1) The authors should perform calibration experiments on humans to clear the concern that the control and Temporal Odor Mixture (TOM) discrimination experiments were different. These differences are both by way of total odor and in the way the delivery apparatus was designed.

2) The authors should extend their longest odor interval from 300 ms to around 600 ms. This would address the concern that their longest tested interval, expressed as a fraction of respiration cycle duration, is actually rather shorter than most experiments in rodents where temporal discrimination has been reported.

In addition, the reviewers have made several other suggestions that the authors may wish to address to improve the manuscript.

Reviewer #1:

In this behavioural study Perl et al. ask whether humans can discriminate between odour pairs delivered in different temporal orders. They find that humans can discriminate if they know that two odour will be given in sequence, but not otherwise. This study adds to the substantial literature on the importance of timing in odour perception.

The authors take some pains to deliver odour precisely with a fixed duration of 200 ms and varying intervals between odour. They also explore a variety of relevant control stimuli in Figure 2. Overall the experiments seem carefully done.

The intriguing finding of this study is that odour order discrimination is possible at 300 ms interval, but only if the subjects know that the stimuli are sequential. The authors argue that if timing were an important component of odour coding then the percepts for different odour should be quite different.

1) I wonder if the authors have narrowed their interval proportionately to the discrimination time in model organisms. The authors begin to see discriminability at 300ms, of a respiration respiration cycle time of about 6 seconds (e.g., Figure 1F). Here a 300 ms interval is 1/20 of the cycle. In various animal studies the separation of stimuli is from 13 ms (Rebello et al) to 50 ms (Cury and Uchida and others). The mouse/rat sniff cycle can be well under 100 ms, though note that the Cury and Uchida study points out timing the initial respiration phase is quite similar even for slower respiration. Thus the phase separation is from 1/8 to 1/2 of a cycle. Thus in this rather rough calculation, one might expect that if human discrimination scales similarly to their respiration rates, we should expect discrimination to start to improve at about 600 ms, which is twice as long as what the authors try. Thus one explanation for their findings may simply be that they used too short an interval.

2) I feel that the study is a little uni-dimensional: here is a human perceptual observation, without mechanistic underpinnings, and only using two pairs of odour. While it does have interesting interpretations in terms of coding as suggested by the authors, it would have been good to have these interpretations backed by further observations within the constraints of human experimentation.

Reviewer #2:

In this manuscript, Perl et al. aim to assess the contribution of glomerular activation timing to the problem of odor identification. To this end, they use temporal odor mixtures (TOMs: A->B vs. B->A, each presented within one sniff) to presumably activate the same set of glomeruli in two sniffs but in different temporal sequences. They ask human participants to report whether the odor percepts across the two sniffs are the 'same' or 'different'.

Perl et al. find that human participants largely fail to discriminate between TOMs (chance performance) while showing significantly higher discrimination performance on simpler discrimination tasks (A vs. B, A->B vs. A or A->B vs. B). However, TOM discrimination does improve when participants are pre-informed about the composition of TOMs as sequences of two different odors. The authors propose that latency differences in timing of glomerular activation may not be automatically perceived as differences in odor identity. However, information pertaining to latency differences can be extracted to enhance odorant discriminability when required.

The degree to which glomerular timing contributes to odor identification is a longstanding question. As the authors themselves note, substantial body of work in non-human systems has shown that small differences in glomerular activation times elicit changes in both neuronal and behavioral responses. However, behavioral results from non-human model systems frequently rely on extensive training, thus making it hard to distinguish between what the olfactory system does versus what the olfactory system can do?

In this context, the results of Perl et al. are extremely relevant: they provide the clearer view of what features of the glomerular activity patterns olfactory system natively tunes to in order to identify odors. Overall, the study is well executed and the results are presented clearly.

However, my enthusiasm to support publication at this stage is dampened by several concerns regarding the interpretation of their results. These should be addressable by one additional set of behavioral experiments (point 1).

Concerns:

The authors' primary conclusion (latency differences in glomerular timing do not contribute to odor identification) rests on one key observation: human subjects show poorer discrimination performance for TOM discrimination versus control conditions.

While the authors do perform a battery of controls, the stimulus properties in the control conditions differ significantly from those in the TOM discrimination experiments. The total amount of odor delivered per sniff and spatial distribution of airflow across the final tubes (see points 1 and 2) differ in the TOM discrimination and in the controls. This makes the results hard to interpret: the deterioration in discrimination performance for TOMs may result from factors other than just differences in glomerular activation times. It is essential to rule out these alternative factors with additional control experiments (see point 1) before publication.

1) The authors' odor discrimination paradigm requires subjects to compare percepts across two sniffs (~8 s apart). In the TOM experiments, the subjects experience odors for a total of 400 ms (200ms of A + 200ms of B) in each sniff. However, in control conditions the subjects experience either overall less amount of odor (odor A vs. B; 200ms in each sniff) or different amounts of odor across the two sniffs (task: A->B vs. A/B->air; 400 ms in the first sniff and 200 ms in the second sniff).

It is unclear how discrimination performance changes with total odor content per sniff. Do the subjects perform poorer in the TOM task simply because of the higher odor content in both the sniffs being compared? Likewise, can the subjects simply rely on the difference in total odor content between the two sniffs to assess stimulus similarity in the second set of controls (A->B vs. A/B->air)?

In my opinion, a more interpretable control is to test whether subjects can discriminate A->B vs. A->A or A->B vs. B->B better than the TOM condition. This would allow the authors to maintain the same odor content across each sniff across both experimental and control conditions.

2) Flow related cues might allow subjects to achieve a higher performance in control conditions compared to the TOM task. The authors use a non-typical final port design for odor delivery, where each stimulus (A, B and air) is delivered through a separate tube. From the Materials and methods section, it is not clear whether the airflow is ON through all three tubes at all times. When odor A is OFF, is the tube dedicated to odor A delivering clean air instead, such as to maintain the same net air flow?

If not, another axis along which subjects can discriminate across stimuli is by comparing spatial distribution of airflow patterns across tubes. The TOM discrimination experiment is the only experiment where the airflow is ON through same set of tubes (tubes A and B) across the two sniffs. For all the control conditions, airflow switches from one set of tubes in the first sniff to another set of tubes in the second sniff. For the first set of controls (A vs. B), the flow switches from tube A to tube B. For the second set of controls (A->B vs. A->air), the flow switches from tube A to the air tube. Given these limitations, it is impossible to rule out that poorer discrimination in the TOM task (compare to controls) simply results from the smaller differences in flow cues across the two sniffs.

3) Discrimination performance varies across control conditions (Figure 2A and 3A) in ways that are not obvious. Naively, it appears, that discriminating A vs. B should be easiest. While this is true for CTL-DMTS odor pair, it does not seem to be the case for the CIN-ORG pair. There are asymmetries in performance across controls that are expected to show similar outcomes. Why is A->B vs. A->air harder to discriminate than A->B vs. B->air?

Since the deterioration in discrimination performance is the metric that the authors base their claims on, the authors should comment on the variations in this metric and the factors that underlie these variations. Are these differences across control conditions significant, especially given that the control conditions consistently have fewer subjects than the TOM experiments? If significant, the authors should comment on possible explanations of these differences in a manner that is consistent with the results observed in the TOM discrimination task.

Reviewer #3:

In this paper, Perl and colleagues attempt to analyze if glomeruli activation times affect odor perception by performing olfactory discrimination experiments with human subjects. Authors designed temporal odor mixtures (TOMs) composed of two components and challenged the subjects to discriminate between the stimuli where the sequences of presentation of these components were different. Results show that the subjects were unable to discriminate these TOMs when they were not informed about the sequential presentation of components. This is an important topic that needs to be discussed in the context of temporal coding in olfaction.

As the authors mention, temporal coding in olfaction has been discussed extensively and given many interpretations. Earlier works have provided strong experimental evidences for relative time-based code in rodent olfactory system (Haddad et al., 2013; Smear et al., 2013). Here authors rely on the behavioral readouts to study how the time-based code contributes to olfactory information processing. While I agree that this is an important topic, lack of clarity with the interpretations and the flaws with the experimental design question the aptness of this article for a publication in *eLife* in the present format. The study needs to be revised.

Here are my major concerns:

1) This study tried to address how temporal coding affects odor perception. To probe this, authors designed the TOMs, made of odors A and B that varied in sequence of presentations. When the sequence of presentation changes from A→B to B→A, the relative time of glomeruli activation changes. I agree with authors' claim of differences in the relative time of glomerular activation for odors A and B separately ("Delivering two odors at different order activates the glomeruli at a different relative time and form a different latency code"). But, are authors neglecting the information processed by odor pulse B, when it is presented first in the sequence in a discrimination context? Is this the optimal stimuli combination for the question they are addressing?

2) In the section of "temporal dynamics can be used to exploit odor related features", authors discuss that glomeruli activation time can be utilized to extract odor-related information such as when the odor was delivered relative to other odors or relative to respiration phase. This can be tested by challenging the subjects with a discrimination task using the same target odor pulse at different timings in the background of another odor. For me, this is more relevant as we have to detect and discriminate specific odors in olfactory enriched environments.

3) What do authors try to address by reducing the time delay between the onset of pulses to 75 ms and 150 ms for CTL/DMTS discrimination? This would allow the mixing of two odors and the subjects could perceive this as a single entity (?).

4) I would like to see a more detailed discussion and comparison between this study and other studies (Haddad et al., 2013, Rebello et al., 2014 and Smear et al., 2013) to draw more robust conclusions about the neural mechanisms involved.

Other concerns:

1) Technical issues: authors have used Orange and Cinnamon extracts. These extracts are mixtures of different monomolecular odors that vary in their physico-chemical properties. Does this affect authors' conclusion?

2) Authors have cited Friedrich and Laurent, 2001 and Gschwend et al., 2015 for the decorrelation model. While I respect authors' freedom to interpret the results differently, these two studies are reporting totally different time scales for the decorrelation (slow vs. fast).

3) Interpretations given for Rebello et al., 2014 and Smear et al., 2013 are misleading.

4) In the last experiment, when participants were informed about the nature of the task, authors observed incremental improvement in the performance. Authors say that it helped in 'unlocking' the discriminability. What is the neural mechanism underlying this "unlocking"?

5) Few p values reported are the same (Subsection “The effect of odor temporal dynamics on odor perception” paragraph three and subsection “The effect of rapid odor temporal dynamics and odor similarity on odor perception” paragraph two).

---

## [Author Response]

The reviewers all agreed that the results were surprising and potentially interesting. They brought up a number of technical and conceptual points that they felt could be cleared up by some rather straightforward experiments of the same design.1) The authors should perform calibration experiments on humans to clear the concern that the control and Temporal Odor Mixture (TOM) discrimination experiments were different. These differences are both by way of total odor and in the way the delivery apparatus was designed.2) The authors should extend their longest odor interval from 300 ms to around 600 ms. This would address the concern that their longest tested interval, expressed as a fraction of respiration cycle duration, is actually rather shorter than most experiments in rodents where temporal discrimination has been reported.In addition, the reviewers have made several other suggestions that the authors may wish to address to improve the manuscript.

We thank all reviewers and the reviewing editor for their helpful and constructive comments.

All reviewers agree that this finding is interesting as it sheds new light on the contribution of latency coding in odor perception in humans. Two technical concerns were raised (as summarized above by the reviewing editor). We have addressed these concerns by conducting the suggested three experiments and conducting new analyses. We have recruited overall additional 59 participants for these three new experiments (reported in Figures 2A, 3A, 4D-F and Figure 4—figure supplement 1).

To summarize, we ran two additional control experiments, specifically designed to mitigate the reviewers’ concerns, namely to address the possibility that the control and the TOMs are different in terms of odor duration and airflows (These are now detailed in Figure 4—figure supplement 1).

We have also expanded the range of our testing by introducing a new, longer, delay of 600 ms between odor constituents within the TOMs, asking whether participants can discriminate between TOMs in terms of accuracy and perception. Notably, this effort is reflected horizontally in essentially all main experiments in the manuscript ms (Discrimination – Figures 2A, 3A, and 5; Perception – Figure 4D-F).

The results of all these three experiments further strengthen the study and we thank the reviewers for the opportunity to clarify these possible concerns.

Finally, we have implemented your comments regarding minor issues of clarity throughout the manuscript and have further elaborated where was required.

Reviewer #1:In this behavioural study Perl et al. ask whether humans can discriminate between odour pairs delivered in different temporal orders. They find that humans can discriminate if they know that two odour will be given in sequence, but not otherwise. This study adds to the substantial literature on the importance of timing in odour perception.The authors take some pains to deliver odour precisely with a fixed duration of 200 ms and varying intervals between odour. They also explore a variety of relevant control stimuli in Figure 2. Overall the experiments seem carefully done.The intriguing finding of this study is that odour order discrimination is possible at 300 ms interval, but only if the subjects know that the stimuli are sequential. The authors argue that if timing were an important component of odour coding then the percepts for different odour should be quite different.1) I wonder if the authors have narrowed their interval proportionately to the discrimination time in model organisms. The authors begin to see discriminability at 300ms, of a respiration respiration cycle time of about 6 seconds (e.g., Figure 1F). Here a 300 ms interval is 1/20 of the cycle. In various animal studies the separation of stimuli is from 13 ms (Rebello et al) to 50 ms (Cury and Uchida and others). The mouse/rat sniff cycle can be well under 100 ms, though note that the Cury and Uchida study points out timing the initial respiration phase is quite similar even for slower respiration. Thus the phase separation is from 1/8 to 1/2 of a cycle. Thus in this rather rough calculation, one might expect that if human discrimination scales similarly to their respiration rates, we should expect discrimination to start to improve at about 600 ms, which is twice as long as what the authors try. Thus one explanation for their findings may simply be that they used too short an interval.

We thank the reviewer for this interesting suggestion. The reviewer is suggesting that temporal coding might be proportional to the sniff length. That is, in mice temporal coding is in the range of 10-50 ms (in Smear et al. they report a temporal resolution of 10ms), while in model organisms possessing longer sniffs this could scale up to be hundreds of milliseconds. We think this is quite an interesting claim. We are not aware of any study suggesting that the scale of temporal coding is relative to the sniff length. Notably, in insects, there is no respiratory cycle per se onto which sniffs can map. Furthermore, if this suggestion is correct then it has deep implications on the general notion of how neurons work: Downstream neurons in humans should possess a very long integration window so that they could be sensitive only to very long temporal differences but not short ones. That said, we extended our study to include a delay of 600 ms and the results are very similar to those observed with 300 ms (now reported in Figure 2A and 3A and Figure 4D-E and Figure 5).

Originally, we did not test a 600 ms delay since it was technically challenging for two reasons: First, with a 600 ms delay, the overall duration of each TOM is 800 ms (200 ms+400 ms+200 ms) which is sometimes at the limit of the inhalation length (~1-1.5 seconds). This resulted in many trials that had to be excluded since our predefined inclusion criterion demanded that the entire odor stimulus has to occur within the inhalatory phase. Second, and more importantly, when the delay is as long as 600 ms it often happens that participants perceived the stimulus as two odors arriving one after the other. This may result in a scenario where we are not testing if participants perceive the TOM as a new odor but rather if they can tell the order of the odor sequences (which is similar to the second set of experiments we conducted, in which we notified the participants in advance each trial is constructed of two odors arriving one after the other).

Nevertheless, to overrule this possible concern, we run another set of experiments in which we tested TOM discrimination in the presence of a 600 ms delay. As expected, we had to exclude quite a few participants in which the TOM wasn’t fully presented during the inhalation period. The results are presented in Figures 2A and 3A. The vast majority of the participants failed to significantly discriminate between the TOMs in an accuracy surpassing chance level. (Figures 2A and 3A in the 600 ms condition). The success rates of the group in the 600 ms delay is similar to the one in 300 ms. In both experiments the success rates are still far lower than the rates obtained in the control experiments or when the participants are aware of the nature of the TOMs (reported in Figures 5A and 5B). As expected, few participants reported realizing the TOM are composed of two consecutive odors. We did not exclude these participants although their success rates were above chance level.

To further examine the effect of a 600 ms delay on odor perception, we asked 9 new participants to rate the TOMs with a 600 ms delay and their components using the same verbal descriptors we used in Figure 4A. These results are reported in text and are now added to Figure 4D-F. This experiment clearly shows that even with a 600 ms delay humans perceive the TOMs as similar and not as two distinct odors as is expected from a temporal code per-se.

Finally, we examined performance in the presence of a 600 ms delay when the participants were aware of the temporal dynamics within the stimulus and observed that in this case they were able to easily discriminate between the TOMs (and even better than all shorter delays previously tested). These results are now reported in text and added to Figures 5A and 5B.

Taken together, these experiments demonstrate that latency coding as defined here does not contribute substantially to odor percept even when the delay is as long as 600 ms.

2) I feel that the study is a little uni-dimensional: here is a human perceptual observation, without mechanistic underpinnings, and only using two pairs of odour. While it does have interesting interpretations in terms of coding as suggested by the authors, it would have been good to have these interpretations backed by further observations within the constraints of human experimentation.

We do understand the reviewer’s concern. However, this is already quite a large study compared to other human psychophysics experiments. Overall, we report results from a cohort of 284 participants. This is a within the upper bounds of cohorts reported in a single study in the field of olfactory psychophysics and perception. Furthermore, in the main paradigms presented here each subject’s performance was derived from 20 trials in most cases (and up to 200 trials per subject in the paradigm which unfolded over days, see Figures 3B-C). This culminates into overall approximately 6000 trials which were analyzed in this study. This is somewhat comparable with high-throughput automated studies conducted in rodents (*1*).

We think using two odor pairs (two familiar and two that are highly dissimilar) is reasonable to convey the main point. The argument is that if temporal coding is as indispensable to perception then any two odor pairs which activate the glomeruli at different orders should elicit a different percept. In the discussion we note that it is possible that under some other circumstances (different odors, timing etc.) latency coding could contribute more to perception. Furthermore, unlike studies in rodents, we took advantage of the ability of humans to report what they smell to directly estimate how TOMs are perceived. We think this approach is an important contribution to the study of temporal coding which gives an additional and unique insight.

Finally, we do provide an interesting observation that once participants are told that the TOMs are composed of two consecutive odors, they do seem to dramatically improve in discriminating between them. We did examine how participants were able to succeed in this experiment, and it turns out they were applying an odor matching procedure of either the first odor or the second. This is similar to the known phenomena that people are much better when they actively look for something they know (e.g. a specific pen on a cluttered table) than when they need to find an unknown pen. We wish we could probe the underlying neural mechanism further as it is very interesting (which also is now on our minds as we plan future projects using this setup) but unfortunately it is not something that we could address in humans with a respectable depth.

Reviewer #2:In this manuscript, Perl et al. aim to assess the contribution of glomerular activation timing to the problem of odor identification. To this end, they use temporal odor mixtures (TOMs: A->B vs. B->A, each presented within one sniff) to presumably activate the same set of glomeruli in two sniffs but in different temporal sequences. They ask human participants to report whether the odor percepts across the two sniffs are the 'same' or 'different'.Perl et al. find that human participants largely fail to discriminate between TOMs (chance performance) while showing significantly higher discrimination performance on simpler discrimination tasks (A vs. B, A->B vs. A or A->B vs. B). However, TOM discrimination does improve when participants are pre-informed about the composition of TOMs as sequences of two different odors. The authors propose that latency differences in timing of glomerular activation may not be automatically perceived as differences in odor identity. However, information pertaining to latency differences can be extracted to enhance odorant discriminability when required.The degree to which glomerular timing contributes to odor identification is a longstanding question. As the authors themselves note, substantial body of work in non-human systems has shown that small differences in glomerular activation times elicit changes in both neuronal and behavioral responses. However, behavioral results from non-human model systems frequently rely on extensive training, thus making it hard to distinguish between what the olfactory system does versus what the olfactory system can do?

We are thankful to the reviewer and their comprehensive comments. The last sentence is an insightful and well phrased way to describe the finding. We added the following sentence to the Discussion:

“Furthermore, behavioral results from non-human model systems frequently rely on extensive training (e.g. thousands of trials of highly motivated animals), thus making it hard to distinguish between what the olfactory system does versus what the olfactory system can do. It is thus possible that although latency coding does not substantially affect odor perception temporal dynamics can be utilized under extreme conditions.”

In this context, the results of Perl et al. are extremely relevant: they provide the clearer view of what features of the glomerular activity patterns olfactory system natively tunes to in order to identify odors. Overall, the study is well executed and the results are presented clearly.However, my enthusiasm to support publication at this stage is dampened by several concerns regarding the interpretation of their results. These should be addressable by one additional set of behavioral experiments (point 1).Concerns:The authors' primary conclusion (latency differences in glomerular timing do not contribute to odor identification) rests on one key observation: human subjects show poorer discrimination performance for TOM discrimination versus control conditions.While the authors do perform a battery of controls, the stimulus properties in the control conditions differ significantly from those in the TOM discrimination experiments. The total amount of odor delivered per sniff and spatial distribution of airflow across the final tubes (see points 1 and 2) differ in the TOM discrimination and in the controls. This makes the results hard to interpret: the deterioration in discrimination performance for TOMs may result from factors other than just differences in glomerular activation times. It is essential to rule out these alternative factors with additional control experiments (see point 1) before publication.1) The authors' odor discrimination paradigm requires subjects to compare percepts across two sniffs (~8 s apart). In the TOM experiments, the subjects experience odors for a total of 400 ms (200ms of A + 200ms of B) in each sniff. However, in control conditions the subjects experience either overall less amount of odor (odor A vs. B; 200ms in each sniff) or different amounts of odor across the two sniffs (task: A->B vs. A/B->air; 400 ms in the first sniff and 200 ms in the second sniff).It is unclear how discrimination performance changes with total odor content per sniff. Do the subjects perform poorer in the TOM task simply because of the higher odor content in both the sniffs being compared? Likewise, can the subjects simply rely on the difference in total odor content between the two sniffs to assess stimulus similarity in the second set of controls (A->B vs. A/B->air)?In my opinion, a more interpretable control is to test whether subjects can discriminate A->B vs. A->A or A->B vs. B->B better than the TOM condition. This would allow the authors to maintain the same odor content across each sniff across both experimental and control conditions.

We thank the reviewer for raising these concerns. This is a valid point which also reflects a comprehensive thinking effort on the reviewer’s part and we are thankful for this observation. With your advice in mind we carried out the exact controls you suggested (A->B vs. A->A and A->B Vs B->B). Performance in these controls, centered around 80% accuracy, indicating that participants can easily discriminate these two set of TOMs (now presented in Figure 4—figure supplement 1). Furthermore, to assert this finding is not due to serendipitous recruiting of high-performing participants, we verified that these same participants cannot discriminate between A->B Vs. B->A, thus forming a matched pair test which is statistically stronger and further strengthen the results. This contrasts between high performance in control and a 50-60% accuracy in main paradigm conveys a clear message – even when equating total odor content and duration, participants cannot discriminate between the TOMs tested here.

2) Flow related cues might allow subjects to achieve a higher performance in control conditions compared to the TOM task. The authors use a non-typical final port design for odor delivery, where each stimulus (A, B and air) is delivered through a separate tube. From the Materials and methods section, it is not clear whether the airflow is ON through all three tubes at all times. When odor A is OFF, is the tube dedicated to odor A delivering clean air instead, such as to maintain the same net air flow?If not, another axis along which subjects can discriminate across stimuli is by comparing spatial distribution of airflow patterns across tubes. The TOM discrimination experiment is the only experiment where the airflow is ON through same set of tubes (tubes A and B) across the two sniffs. For all the control conditions, airflow switches from one set of tubes in the first sniff to another set of tubes in the second sniff. For the first set of controls (A vs. B), the flow switches from tube A to tube B. For the second set of controls (A->B vs. A->air), the flow switches from tube A to the air tube. Given these limitations, it is impossible to rule out that poorer discrimination in the TOM task (compare to controls) simply results from the smaller differences in flow cues across the two sniffs.

The reviewer raises the possible concern that perhaps participants could more easily discriminate between A→B and A→C because in this experiment, different tubes are involved in each TOM and therefore participants could have exploited some changes in airflow (or spatial cues).

First, we think it is unlikely that these differences in success rates can be explained solely by the changes in airflow. The setup is built such that all odor ports and air ports are exactly the same (same bottle types, same connectors, same connecting tubes and valves etc.) so the flow should be the same across ports and was verified to be the same using a flow meter (Alicat Scientific Flow meter).

That said, in order to address this concern directly, we ran an additional control experiment in which we repeated the A→B Vs B→A experiment, but this time we used the ports previously designated as C for the B odor when we delivered the B→A TOM (we call it C(B)→A). C(B) delivers odor B through the C stream and therefore this experiment resembles the A→B Vs. A→C experiment in terms of changes in airflow while still delivering the same odors: A→B and B→A. If the airflow (or any other difference in the C port) provides information to help participants distinguish between AB and AC then it should also allow distinguishing between AB and B(C)A. We found that all participants but one failed to discriminate between A→B and C(B)→A indicating that changes in airflow don’t provide sufficient cues to help in discriminating between the two TOMs (Figure 4—figure supplement 1). Interestingly, the one participant who did succeed reported perceiving that the TOM is composed of two odors arriving one after the other which, for this individual, renders this experiment similar to the ones we report in Figure 5 in which participants easily discriminated between the TOMs when we have notified them in advance that the TOMs are composed of two consecutive odors.

3) Discrimination performance varies across control conditions (Figure 2A and 3A) in ways that are not obvious. Naively, it appears, that discriminating A vs. B should be easiest. While this is true for CTL-DMTS odor pair, it does not seem to be the case for the CIN-ORG pair. There are asymmetries in performance across controls that are expected to show similar outcomes. Why is A->B vs. A->air harder to discriminate than A->B vs. B->air?Since the deterioration in discrimination performance is the metric that the authors base their claims on, the authors should comment on the variations in this metric and the factors that underlie these variations. Are these differences across control conditions significant, especially given that the control conditions consistently have fewer subjects than the TOM experiments? If significant, the authors should comment on possible explanations of these differences in a manner that is consistent with the results observed in the TOM discrimination task.

We thank the reviewer for this important comment and we have now explicitly addressed it in the manuscript. We think the reason for the higher success rate in discriminating between the CTL-DMTS is because these two odors are far more dissimilar than the CIN-ORG group. We now added an analysis in which we directly assess the perceptual distances between these odor pairs. CIN and ORG are two pleasant smells and as such it is not as easy to discriminate as DMTS (an unpleasant odor reminiscent of rotten egg) and CTL (a generally perceived as pleasant odor reminiscent of an orange).

It is possible that AB/AC is slightly harder to discriminate than AB/BC in the CIN-ORG experiment, because maybe A, as the first odor in the sequence, is slightly more dominant at least for some of the participants causing AB to be more similar to AC but not to BC. However, it is important to note that the success rates of all control experiments are not significantly different as stated: “…, all p’s of cross-control comparisons > 0.68)”, and therefore, it is possible that these differences are just statistical fluctuations caused by the selection of some less competent participants and because the group size may not be large enough to mitigate such fluctuations. Importantly, this difference does not occur in the CTL-DMTS pair.

The difference in success rates between AB/AC compared to A/B is expected as AB/AC share an odor component and therefore by definition are more similar.

Reviewer #3:In this paper, Perl and colleagues attempt to analyze if glomeruli activation times affect odor perception by performing olfactory discrimination experiments with human subjects. Authors designed temporal odor mixtures (TOMs) composed of two components and challenged the subjects to discriminate between the stimuli where the sequences of presentation of these components were different. Results show that the subjects were unable to discriminate these TOMs when they were not informed about the sequential presentation of components. This is an important topic that needs to be discussed in the context of temporal coding in olfaction.As the authors mention, temporal coding in olfaction has been discussed extensively and given many interpretations. Earlier works have provided strong experimental evidences for relative time-based code in rodent olfactory system (Haddad et al., 2013; Smear et al., 2013). Here authors rely on the behavioral readouts to study how the time-based code contributes to olfactory information processing. While I agree that this is an important topic, lack of clarity with the interpretations and the flaws with the experimental design question the aptness of this article for a publication in eLife in the present format. The study needs to be revised.Here are my major concerns:1) This study tried to address how temporal coding affects odor perception. To probe this, authors designed the TOMs, made of odors A and B that varied in sequence of presentations. When the sequence of presentation changes from A→B to B→A, the relative time of glomeruli activation changes. I agree with authors' claim of differences in the relative time of glomerular activation for odors A and B separately ("Delivering two odors at different order activates the glomeruli at a different relative time and form a different latency code"). But, are authors neglecting the information processed by odor pulse B, when it is presented first in the sequence in a discrimination context? Is this the optimal stimuli combination for the question they are addressing?

We thank the reviewer for this comment which reflects a need for clarification on our side. Our initial finding with the odors of orange and cinnamon was indeed tested only with the TOMs of AB Vs. AB / BA. Following coming across the main finding of this study (namely the poor discrimination ability) and the application of adequate controls (Figure 2) we have next tested both A->B and B->A as the first stimuli within the TOM in ensuing experiments (see Figure 3 and its legend). Thus, both options are covered. We hope this addresses the reviewer concern.

2) In the section of "temporal dynamics can be used to exploit odor related features", authors discuss that glomeruli activation time can be utilized to extract odor-related information such as when the odor was delivered relative to other odors or relative to respiration phase. This can be tested by challenging the subjects with a discrimination task using the same target odor pulse at different timings in the background of another odor. For me, this is more relevant as we have to detect and discriminate specific odors in olfactory enriched environments.

We thank the reviewer for this comment. The concept of presenting an odor pulse in the background is indeed an elegant way to probe discrimination through target-background stimulus extraction. This exact method was employed before to study adaptation in human olfaction by David Smith (see Smith et al., 2010).

We found this design to be very different from the one used in our study and therefore we did not refer to it, let alone attempted to replicate its methodology.

We now acknowledge that some may find this study to be of relevance in the context of interpretation of our results and therefore we added the following text to the Discussion:

“Few participants reported they were actively searching for a specific odor to occur at the beginning or end of the stimulus, suggesting they employ a matching algorithm for one of the constituents. When the delay was set to be shorter than 150 ms, this matching failed because the first odor was presented in partial temporal overlap with the second one. This strategy is also in line with a previous study in which participants were able to name which odor out of two known odors was presented first when the delay was 200-400 ms (Laing et al., 1994) or when it is presented in the presence of a background stimulus (Smith et al., 2010).”

3) What do authors try to address by reducing the time delay between the onset of pulses to 75 ms and 150 ms for CTL/DMTS discrimination? This would allow the mixing of two odors and the subjects could perceive this as a single entity (?).

We thank the reviewer for this question. We did not want to base all our finding on a single parameter for delay between TOMs. We therefore explored other delays as well. Initially we did not want to go above 300 ms for reasons detailed in the manuscript as well as within this response letter. Previous experiments found that mice can discriminate optogenetic stimulations when the delay is in the order of few tens of milliseconds. We therefore wanted to examine if temporal coding might be more effective in shorter delays. Furthermore, reducing the delay offered some insights as to a (somewhat crude) performance curve which offered a richer understanding of the phenomenon we set out to study.

Please see our reply to your first comment in the “Other concerns” section just below, regarding the question of a potential “mixing of two odors into a single entity”.

Finally, as also suggested by the first reviewer, we have now extended the study to include long delays of 600 ms. The results further support the main claim of this study.

4) I would like to see a more detailed discussion and comparison between this study and other studies (Haddad et al., 2013, Rebello et al., 2014 and Smear et al., 2013) to draw more robust conclusions about the neural mechanisms involved.

We thank the reviewer for this suggestion. We have now completely revised the first paragraph in the Discussion regarding possible neural mechanisms and comparisons with these studies.

"Odor-elicited temporal dynamics not directly related to stimulus dynamics have been long observed in the olfactory system. These observations gave rise to the hypothesis that odors are encoded by a spatiotemporal code. What does a temporal code mean in the context of the olfactory system has been given several interpretations; one prominent model suggests that the time of activated glomeruli relative to some internal or external event is part of the odor code. In the current study, we set out to test whether the time of glomeruli activation affects odor perception as expected from this model. We used odor stimuli composed of two odors sequentially presented at different orders (TOMs) and observed that the greater majority of participants could poorly discriminate between TOMs (Figures 2A, 3A and 3B)."

And:

“Due to the considerable challenge posed by delivering two odors within the short inhalation time of mice (typically <100 ms in actively smelling mice), the authors conducted these experiments using optogenetic stimulations in mice expressing Channelrhodopsin2 in the olfactory sensory neurons (OSN) or the M/T cells. Mice learned to discriminate between two light stimulations of the OSNs, M/Ts or even a single glomerulus that are few milliseconds apart, either relative to-, or irrespective of respiration (Rebello et al., 2014; Smear et al., 2011, 2013). These experiments clearly demonstrated the use of timing in the olfactory system of mice. However, they did not demonstrate that mice perceived the optogenetic stimulations as two different odor percepts as mice cannot report their odor perception directly. One possible interpretation is that the trained mice perceived the two optogenetic stimuli as the same odor presented at different respiration phases. Alternatively, mice may have perceived the optogenetic stimuli as two different “odor” durations or two odors arriving one after the other. Furthermore, behavioral results from non-human model systems frequently rely on extensive training (e.g., thousands of trials of highly motivated animals), thus making it hard to distinguish between what the olfactory system does versus what the olfactory system can do. It is thus possible that although latency coding does not substantially affect odor perception, temporal dynamics can be utilized under extreme conditions. This may be achieved with the help of neurons that are sensitive to differences in M/T relative activation times (Haddad et al., 2013).”

Other concerns:1) Technical issues: authors have used Orange and Cinnamon extracts. These extracts are mixtures of different monomolecular odors that vary in their physico-chemical properties. Does this affect authors' conclusion?

We thank the reviewer for this comment which reflects a comprehensive grasp of our methodology. The way humans perceive (and probably non-human as well) olfactory stimuli (often referred to as “olfactory objects”) is far from clear. It is largely held that people, especially non-professional, naïve participants, perceive odors synthetically. That is, they cannot tell if an odor is a mono-molecule or a mixture of odorants. See a host of studies by D. G. Laing from the late 90’s which provided mounting evidence to back up this claim as well as a recent review by Benjamin D. Young (Mind and Language, 2019) who summarized those to claim that “The inability to perceptually identify the constituents within a complex smell is best explained in light of the aforementioned evidence that the nature of sensory and cortical encoding of olfactory stimuli does not always encode complex odors as the concatenation of their constituents”.

We therefore believe our approach presenting TOMs of mixtures of Orange and Cinnamon smells is valid.

Furthermore, we verified that both mixtures are perceived as equally intense and outsourced them to a gas-chromatography mass-spectrometry analysis for a breakdown of their components which is listed in the Materials and methods section of the manuscript under “experimental design”.

Finally, as noted in the manuscript, we had several reasons to conduct the next experiments with a newly introduced odor pair of CTL and DMTS, both of which are mono-molecules. One of the reasons was to move into “cleaner” perception. As we detail in the manuscript, both types of odor pairs resulted in poor discrimination ability between temporal sequences of odors.

2) Authors have cited Friedrich and Laurent, 2001 and Gschwend et al., 2015 for the decorrelation model. While I respect authors' freedom to interpret the results differently, these two studies are reporting totally different time scales for the decorrelation (slow vs. fast).

We concur. We removed the Gschwend et al. citation from this section.

3) Interpretations given for Rebello et al., 2014 and Smear et al., 2013 are misleading.

We absolutely agree with the reviewer. We removed this paragraph.

4) In the last experiment, when participants were informed about the nature of the task, authors observed incremental improvement in the performance. Authors say that it helped in 'unlocking' the discriminability. What is the neural mechanism underlying this "unlocking"?

We thank the reviewer for this question. Unlocking is a very fitting word to describe a sudden improvement in perceptual abilities, yet it conveys very little as to the underlying Mechanism. In the discussion we speculated it might be related to pattern matching mechanisms similar to the way one can easily find a pen on a cluttered table when he/she knows how it looks like and it takes much more time to find it when you are just looking for a general pen. To reveal any neural mechanism underlying this phenomenon would necessitate recording of neurons of olfactory circuits which is not feasible with this human cohort and is out of the scope of this study. We do agree this is a highly intriguing question which worth pursuing with the right tools. For human experimentation, probing modulation of attention to odor patterns was studied before using fMRI (see studies by Zelano et al., 2011 Neuron and Plailly et al. 2008, JNS) however to the best of our knowledge, never with TOMs.

We added to the discussion the following sentences:

“This may suggest that although latency code only weakly affects the odor percept of the TOM, it could be used to untangle the constituting odors, provided that the sequential nature of the stimuli was revealed and the temporal dynamics do not evolve too rapidly. One possible interpretation of this result is that when a delay was introduced, there is a substantial duration of time in which there is no odor at all. This break between the two odors might be used to detect the existence of two odors and therefore to aid in perceiving them as two odors arriving one after the other. Another possible explanation is that participants performed well because they employed a pattern matching algorithm. Few participants reported they were actively searching for a specific odor to occur at the beginning or end of the stimulus, suggesting they employ a matching algorithm for one of the constituents. When the delay was set to be shorter than 150 ms, this matching failed because the first odor was presented in partial temporal overlap with the second one.”

5) Few p values reported are the same (Subsection “The effect of odor temporal dynamics on odor perception” paragraph three and subsection “The effect of rapid odor temporal dynamics and odor similarity on odor perception” paragraph two).

This is common in a ranked signed test as it compares ranks and the specific value of each data point does not affect the computation (sign-test considers their ranks and not their nominal values and therefore the same p value will be obtained if the number of points and their ranks is the same).